# Observed and modeled Arctic airmass transformations during warm air intrusions and cold air outbreaks

Manfred Wendisch<sup>1</sup>, Benjamin Kirbus<sup>1,2</sup>, Davide Ori<sup>3</sup>, Matthew D. Shupe<sup>4,5,6</sup>, Susanne Crewell<sup>3</sup>, Harald Sodemann<sup>7,8</sup>, and Vera Schemann<sup>3</sup>

**Correspondence:** Manfred Wendisch (m.wendisch@uni-leipzig.de)

Abstract. Profiles of thermodynamic and cloud properties and their transformations during Arctic Warm Air Intrusions (WAIs) and Cold Air Outbreaks (CAOs) were observed during an aircraft campaign, and simulated using the ICON weather prediction model. The data were collected along flight patterns aimed at sampling the same air parcels multiple times, enabling Eulerian and quasi-Lagrangian measurement-model comparisons and model process studies. Within the Eulerian framework, the temperature profiles agreed well with the ICON output although a small model bias of –0.9 K was detected over sea ice during CAOs. Also, the air parcels did not adjust to the changing surface skin temperature quickly enough. The specific humidity profiles were reproduced by ICON with mean deviations of 6.0 % and 19.5 % for WAIs and CAOs, respectively. Radar reflectivities based on ICON output captured the vertical cloud distributions during the airmass transformations. The simulated process rates of temperature and humidity along the trajectories showed that adiabatic processes dominated the heating and cooling of the air parcels over diabatic effects during both WAIs and CAOs. Of the diabatic processes, latent heating and turbulence had a stronger impact on the temperature process rates than terrestrial radiative effects, especially over the warm ocean surface during CAOs. Finally, a quasi-Lagrangian observation-model comparison was performed. For WAIs, the observed change rates of temperature and humidity were not perfectly captured in the simulations. For the CAOs, the calculated heating and moistening change rates of the airmasses were well represented by ICON with remaining deviations close to the surface.

#### 15 1 Introduction

The recently observed Arctic climate changes have been documented extensively in the international literature (Overland et al., 2011; Jeffries et al., 2013; Richter-Menge et al., 2019). One of the most obvious signs of these changes is the almost 50% decline of the Arctic sea ice extent detected in the time series of the monthly averaged September data since the 1970s (Stroeve et al., 2007; Olonscheck et al., 2019; Serreze and Meier, 2019; Screen, 2021), with a trend of  $-(11.8\pm1.3)\%$  per decade

<sup>&</sup>lt;sup>1</sup>Leipziger Institut für Meteorologie, Universität Leipzig, Leipzig, Germany

<sup>&</sup>lt;sup>2</sup>Now at: Fraunhofer-Institut für Energiewirtschaft und Energiesystemtechnik, Kassel, Germany

<sup>&</sup>lt;sup>3</sup>Institut für Geophysik und Meteorologie, Universität zu Köln, Cologne, Germany

<sup>&</sup>lt;sup>4</sup>Cooperative Institute for Research in Environmental Sciences, University of Colorado Boulder, Boulder, CO, USA

<sup>&</sup>lt;sup>5</sup>National Snow and Ice Data Center, University of Colorado Boulder, Boulder, CO, USA

<sup>&</sup>lt;sup>6</sup>Physical Sciences Laboratory, National Oceanic and Atmospheric Administration, Boulder, CO, USA

<sup>&</sup>lt;sup>7</sup>Geophysical Institute, University of Bergen, Bergen, Norway

<sup>&</sup>lt;sup>8</sup>Bjerknes Centre for Climate Research, Bergen, Norway

for the years between 1979-2023 (https://www.meereisportal.de/en/maps-graphics/sea-ice-trends#gallery-1). Furthermore, the near-surface air temperature has risen sharply in the Arctic within the last few decades (Serreze et al., 2009; Bekryaev et al., 2010; Wang et al., 2016; Rantanen et al., 2022; Wendisch et al., 2023a). However, since 2012, Arctic warming and the decline of sea ice extent appear to have slowed, particularly in winter (Neng et al., 2025). The processes and feedback mechanisms behind these ongoing Arctic climate changes are summarized under the term of Arctic amplification (Serreze and Francis, 2006; Serreze and Barry, 2011). Major observational campaigns have been conducted to disentangle the main reasons of changes of the Arctic climate system and the important factors driving Arctic amplification (Uttal et al., 2002; Wendisch et al., 2019; Shupe et al., 2022; Wendisch et al., 2024). Furthermore, model comparisons have been performed to test the ability of numerical models to predict the main features of Arctic weather and climate (Smith et al., 2019; Solomon et al., 2023). Although these efforts have helped to achieve much progress in understanding Arctic amplification (Previdi et al., 2021; Wendisch et al., 2023a), there is still a lack of appropriate observational data to resolve remaining knowledge gaps and thereby improve modeling of the complex Arctic climate system.

One of these issues concerns the model description of reciprocal linkages between Arctic amplification and mid-latitude weather and climate (Ding et al., 2024). These connections are often realized by episodic, poleward-directed inflows of moist and warm air masses from the mid-latitudes into the Arctic, so-called Warm Air Intrusions (WAIs), or the sporadic outflow of dry and cold airmasses from the Arctic into the mid-latitudes (Cold Air Outbreaks, CAOs¹) (Pithan et al., 2018). For example, for CAOs it has been debated whether the changing Arctic climate is linked to extreme weather in North America and Europe (Cohen et al., 2014, 2020). In general, it is unclear how well airmass transformations occurring during WAIs and CAOs are predicted by numerical models.

To resolve these problems, specific processes that could link the Arctic with mid-latitude weather extremes via WAIs and CAOs have been investigated. Numerous individual case studies of WAIs have been evaluated (Tjernström et al., 2019; Ali and Pithan, 2020; You et al., 2021a, b; Svensson et al., 2023; Kirbus et al., 2023), identifying a variety of key aspects. To name just a few examples: The moisture transported into the Arctic associated with WAIs influences clouds and, as a consequence, modifies precipitation formation (Bintanja et al., 2020; Dimitrelos et al., 2020; Viceto et al., 2022; Lauer et al., 2023; Dimitrelos et al., 2023). It has also been shown that WAIs significantly impact the near-surface energy budget in the Arctic (You et al., 2022; Wendisch et al., 2023b). Furthermore, WAIs transport not only heat and moisture but also aerosol particles to the Arctic (Dada et al., 2022), which can influence the development of the microphysical and radiation-related properties of clouds and thus also precipitation (Bossioli et al., 2021).

When WAIs are confined to narrow and elongated moist filaments, they are called Atmospheric Rivers (ARs) (Zhu and Newell, 1998; Gimeno et al., 2014; Nash et al., 2018; Ma et al., 2021). The occurrence of WAIs is investigated by Dufour et al. (2016), and is expected to increase in the future (Bintanja et al., 2020). Kolbe et al. (2023) reports that the increased poleward moisture transport is likely to be caused almost exclusively by ARs. More ARs may increase sea ice loss (Woods and Caballero, 2016; Komatsu et al., 2018; Zhang et al., 2023) and can promote the melting of the Greenland ice sheet (Mattingly et al., 2018).

<sup>&</sup>lt;sup>1</sup>This study is limited to marine CAOs

CAOs were also investigated by dedicated observational campaigns (Hartmann et al., 1997; Brümmer and Thiemann, 2002; Vihma et al., 2003; Lüpkes et al., 2012; Chechin et al., 2013; Geerts et al., 2022; Kirbus et al., 2024). The most intense CAOs occur in winter (Fletcher et al., 2016; Dahlke et al., 2022) due to the strong thermal contrast between frozen and unfrozen ocean surfaces at that time of year. It is expected that the number of CAOs in winter decreases in the future (Landgren et al., 2019). At the beginning of their development, when the cold airmasses leaving the Arctic sea ice move over the relatively warm open ocean surface, strong airmass transformations occur because of large surface energy fluxes of sensible and latent heat. These energy fluxes can exceed 500 W m<sup>-2</sup> (Tetzlaff et al., 2015; Papritz and Spengler, 2017), which can cause the near-surface air temperature to rise by more than 20 K in only a few hours (Pithan et al., 2018; Wendisch et al., 2023b).

While Atmospheric Boundary Layer (ABL) processes are essential for airmass transformations, model comparisons suggest that there are significant issues representing vertical temperature and humidity profiles, especially with regard to frequent severe temperature inversions near the surface (Pithan et al., 2016). In a related sense, the representation of cloud radiative effects, atmospheric mixing, and atmospheric energy fluxes present further challenges (Kretzschmar et al., 2020; Solomon et al., 2023). A detailed study with individual tendency output showed that during CAOs, large rates of change of different parameterized processes compensate one another, thereby contributing to model uncertainty (Kähnert et al., 2021). In addition to these modeling problems, there is still a general lack of observational data that could be used to assess the spatial-temporal evolution of the properties of cloudy air masses during synoptic transport events, especially near the ground.

To capture Arctic airmass transformations using models and measurements, single-column modeling of Lagrangian airmass changes (Karalis et al., 2025) and a novel quasi-Lagrangian approach have been realized within the HALO– $(\mathcal{AC})^3$  aircraft campaign performed in March and April 2022 (Wendisch et al., 2021, 2024; Walbröl et al., 2024; Ehrlich et al., 2025). The acronym HALO stands for High Altitude and Long Range Research Aircraft (https://www.halo-spp.de/).  $(\mathcal{AC})^3$  indicates a project named " $\mathcal{A}$ rcti $\mathcal{C}$   $\mathcal{A}$ mplification:  $\mathcal{C}$ limate Relevant  $\mathcal{A}$ tmospheric and Surfa $\mathcal{C}$ e Processes and Feedback Mechanisms" (https://www.ac3-tr.de/).

70

85

HALO– $(\mathcal{AC})^3$  delivered numerous observations of thermodynamic and cloud properties along pronounced WAIs and CAOs over open ocean and sea ice, which have been introduced and summarized by Wendisch et al. (2024). This publication also motivated extensively the general need for a Lagrangian-based model evaluation and the required quasi-Lagrangian observations, including their practical realization by aircraft measurements. In the current study, we go one step beyond by exploiting the HALO– $(\mathcal{AC})^3$  measurements in synergy with simulations conducted with the ICON (Icosahedral Nonhydrostatic) weather forecast model to investigate airmass transformations during WAIs and CAOs. For this purpose, we pursue three objectives in this paper:

- Objective 1: We test the ability of the ICON model to reproduce measurements of vertical profile of thermodynamic and cloud quantities from dropsondes and cloud radar in an Eulerian framework. First, two specific cases are used to showcase our approach: a massive WAI (13 March 2022), and a pronounced CAO (01 April 2022). Secondly, the Eulerian measurement-model comparisons are extended to results from further cases from flights over the entire measurement period (six days with WAIs, six days with CAOs).

- Objective 2: We exploit the ICON simulations to investigate the thermodynamic and cloud evolution of the airmasses along their trajectories. This enables to study the role of adiabatic versus diabatic processes for temperature changes, which is further refined to the specific diabatic effects of radiation, latent heat, and turbulence.
- Objective 3: We conduct a novel quasi-Lagrangian model evaluation by testing how well the ICON model simulates measured heating and cooling rates (temperature change rates), as well as moistening and drying rates (humidity change rates).

This article is structured in six sections. After the introduction (Section 1), Section 2 describes the simulations, measurements, as well as the Eulerian and quasi-Lagrangian sampling strategies applied in this study. As the quasi-Lagrangian approach heavily relies on the quality of trajectories, their quality is assessed in Appendix A. The three main parts (Sections 3, 4,
and 5) address the three objectives of the paper. They contain the Eulerian comparisons of ICON model results with aircraft
observations collected during WAIs and CAOs (Section 3, Objective 1), and the discussion of modeled airmass transformations
and processes driving them (Section 4, Objective 2). Section 5 discusses ICON model results and the corresponding measurements quantifying the temperature and humidity change rates during transport of airmasses (Objective 3). The final part of this
paper (Section 6) summarizes the discussion and concludes the article.

# 2 Data and Methods

#### 2.1 Simulations

105

110

115

90

The temporal evolution of the atmospheric state variables, energy and mass fluxes, as well as process tendencies was simulated for each research flight of the HALO– $(\mathcal{AC})^3$  campaign using the ICON model in a limited-area configuration (Zängl et al., 2015). The model domain covered an area from 70 °N to 85 °N, and between 20 °W to 30 °E with a nominal horizontal resolution of 2.4 km. This area contained most of the HALO flight paths during the HALO– $(\mathcal{AC})^3$  campaign (Fig. 1). The atmosphere was discretized along the vertical dimension by 150 terrain-following height levels with a variable resolution of about 20 m close to the surface to about 400 m at the domain top, which was set to 21 km above mean sea level. The initial and lateral boundary conditions were interpolated from the operational global ICON model forecasts by the German Weather Service at a nominal resolution of 13 km. Radiative energy flux densities were parametrized by the ecRad module (Hogan and Bozzo, 2018), while the cloud processes were governed by a bulk, single-moment, five-class microphysical scheme. The ICON model was initialized every flight day at 00 UTC and run for 30 forecast-hours with a time resolution of 10 seconds. With typical aircraft take-off times around 9:00 UTC and nine hours flight durations we considered forecasts with lead times between 9 and 18 hours.

The model output for the full three-dimensional (3D) domain was saved with an hourly frequency. The output quantities included the atmospheric state variables such as air temperature, pressure, as well as specific humidity and mass concentrations of the five hydrometeor classes (cloud and ice water, graupel, snow and rain) and the 3D wind vector components. Also,

quantities such as energy and mass fluxes were stored. For the analysis of the physical drivers of airmass transformations (objective 2), the tendencies for temperature and moisture of the individual processes, e.g., radiation, turbulence, were saved.

While the full model output was only available hourly, radar reflectivities were simulated online using the YAC coupler (Hanke et al., 2016) implemented in ICON, providing atmospheric and hydrometer profiles along the aircraft flight track at the model time resolution. These data were used by the Passive and Active Microwave Radiative TRAnsfer (PAMTRA) tool (Mech et al., 2020) to simulate the airborne radar observations along the HALO flight paths. PAMTRA solved the radar equation considering the backscattering properties of cloud particles and the signal attenuation from hydrometeors and atmospheric gases. Herein, assumptions on size, shape, and density of the hydrometeors consistent with the microphysical scheme were made. The scattering and absorption properties of cloud particles were derived from Mie theory for spherical targets for the liquid hydrometeor classes and graupel, while for snow and ice crystals, the Self-Similar Rayleigh-Gans Approximation was employed (Ori et al., 2021). The PAMTRA output has a temporal resolution of 1 min along the flight track, and has the vertical resolution of the ICON model.

### 2.2 Measurements






During the campaign, HALO was based in Kiruna (Northern Sweden;  $67.85\,^{\circ}$ N,  $20.22\,^{\circ}$ E). Several remote sensing instruments mounted on the aircraft such as microwave radiometers, cloud radar, lidar, and radiation sensors (Stevens et al., 2019) delivered a wealth of data. More than 300 dropsondes were launched from HALO during HALO– $(\mathcal{AC})^3$ . Here, we focus on the dropsonde measurements for the thermodynamic profiles (air temperature, T, equivalent potential temperature,  $\theta_{\rm e}$ , specific air humidity, q, relative air humidity, RH) and cloud information from radar reflectivity (Ze) profiles measured by the 35 GHz Doppler cloud radar. The deployed RD41 dropsondes measured air pressure (accuracy:  $0.4\,\mathrm{hPa}$ ), T (accuracy:  $0.1\,\mathrm{K}$ ), RH (accuracy:  $2\,\%$ ), as well as horizontal wind speeds derived from a Global Positioning System (GPS) receiver (accuracy:  $0.2\,\mathrm{m\,s^{-1}}$ ) (Vaisala, 2020; Ehrlich et al., 2025).  $\theta_{\rm e}$  and q were derived from the measured parameters. The radar measurements were processed to a 30 m vertical grid with a sensitivity limit of about  $-40\,\mathrm{dBZ}$  (Ewald et al., 2019).

HALO conducted 17 research flights during the period between 12 March and 12 April 2022, partly in coordination with four other aircraft. Based on forecasts, the paths of all flights were planned such that as many as possible air parcels were matched at multiple points along their trajectories by the HALO observations, which enabled a quasi-Lagrangian tracking of air masses. In this paper, we have investigated measurements from a subset of 12 HALO research flights observing WAIs and CAOs (Fig. 1). For details of the measurement strategy and the whole data set obtained by multiple aircraft during the HALO– $(\mathcal{AC})^3$  campaign the reader is referred to a set of overview papers (Wendisch et al., 2024; Walbröl et al., 2024; Ehrlich et al., 2025).

In our analysis, we highlight two case studies in detail. The HALO flight patterns for these two cases and the locations where dropsondes were launched are illustrated in Fig. 2. Both flights included measurements over sea ice and open ocean.

- Case 1 (13 March 2022, WAI): The HALO flight conducted on this day surveyed an intense WAI with a northward-directed integrated water vapor transport (IVT) of more than 200 kg m<sup>-1</sup> s<sup>-1</sup>. The flight transected through the core of

this WAI at around 75 °N in the Fram Strait until crossing the sea ice edge and continued northward with a total of seven transects of the moist airmass. At about 85 °N, the aircraft turned south and flew back along the intrusions' main axis (Fig. 2a). Twenty-one dropsondes were released during this flight, from which 20 dropsondes were used in our analysis.

- Case 2 (01 April 2022, CAO): On this day, a strong CAO was probed in the Fram Strait north-west of Svalbard (Fig. 2b), with a flight path that featured multiple legs orthogonal to the main flow covering different distances the airmass passed on its way to the south. Forty-one dropsondes were released from HALO during this research flight, and all of them were used in the comparisons of observations with simulations along the flight track.

Figure 1. Geographical map showing the subset of 12 HALO flight paths conducted in the framework of the HALO– $(\mathcal{AC})^3$  campaign that are analyzed in this paper. Six WAIs (panel (a), red lines, 12, 13, 14, 15, 16, 20 March 2022) and six CAOs (panel (b), blue lines, 21, 28, 29, 30 March 2022 and 01, 04 April 2022) are investigated. Full diamonds indicate the location where dropsondes were launched from HALO. During the six WAIs, a total of 114 dropsondes were successfully released; during the six CAOs, overall 133 dropsondes were launched from HALO. The horizontal projection of the drift of the dropsondes (drift distance) between their launch from HALO and the moment they hit the surface was mostly within a 30 km; for the CAO cases the drift distance was mostly much lower (not shown). The background color (blue to white) depicts the mean sea ice concentration during the campaign taken from ERA5 reanalysis data.

# 2.3 Eulerian and quasi-Lagrangian sampling strategies


To address Objective 1 of this study, we applied the classical Eulerian perspective. We compared profile measurements of thermodynamic quantities  $(T, \theta_e, q, RH)$  from dropsondes and of radar reflectivities (Ze) from cloud radar with their model counterparts. For this purpose, we extracted the simulated profiles closest to the measurement in space and time whereby we

Figure 2. Geographical maps of the HALO flight paths conducted during two case studies in the framework of HALO– $(\mathcal{AC})^3$ . Diamonds indicate the location where dropsondes were released from HALO, whereby the colors mark the temporal distance (in hours) the air parcel travels with the local wind field from the location where the sonde was launched to the 50 % sea ice cover line (Marginal sea Ice Zone, MIZ). If the temporal distance is negative then the air parcel at the location of dropsonde release needs time to reach the MIZ (air parcel moving towards MIZ). If the temporal distance is positive then the air parcel has passed the MIZ already (air parcel moving away from MIZ). (a) HALO flight track (light blue line) covering a WAI on 13 March 2022. The background colored area depicts the integrated water vapor transport, IVT in Fig. 6a of Walbröl et al. (2024), derived from ERA5 reanalysis data of this day at 12 UTC. (b) The light red line indicates the flight path of HALO observing a CAO on 01 April 2022. The 12 UTC ERA5 winds at 0.1 km altitude above ground are shown as barbs. The colored background indicates the CAO index, M in Fig. 6b of Walbröl et al. (2024), calculated from ERA5 data. In both panels (a) and (b), the light (dark) gray solid isolines depict the 20 % (80 %) sea ice concentration retrieved from ERA5.

referred to the lowest altitude of the corresponding dropsonde or radar sounding. In this way, it was assured that the difference between the times and locations of the samplings and simulations was small within the ABL where most of the interactions with the underlying surface occur. Please note that the horizontal drift of the dropsondes during their vertical fall, which was always less than 30 km from release at HALO flight altitude to touchdown on the ground, was not taken into account. Considering the horizontal wind speeds, which were generally below  $25 \, \mathrm{m \, s^{-1}}$  (Fig. A1), and the typical dropsonde descent rate of  $11 \, \mathrm{m \, s^{-1}}$  (Vaisala, 2020), a vertical fall of 1 km takes the dropsonde around 90 seconds. This corresponds to a maximum horizontal drift of 2.3 km, which is slightly less than the width of one ICON model grid cell (2.4 km). If the dropsonde falls 2 km vertically, it drifts horizontally through only two grid cells, which should not significantly bias the Eulerian measurement-model comparison. Furthermore, the hourly model output was linearly interpolated to 1 min resolution, to match the temporal resolution of the PAMTRA simulations and to be much closer in time to the measurements.


To investigate airmass transformations in detail (Objectives 2 and 3), a strictly Lagrangian approach would be desired, wherein the coordinate system moves jointly with the corresponding air parcel (also called intrinsic or natural coordinate system). Since the aircraft flies much faster than air parcels move, truly Lagrangian observations are impossible from fast-flying aircraft. Instead, we have designed flight paths aiming to encounter the same air parcel multiple times during one flight or in the course of two consecutive flights. We call this strategy a quasi-Lagrangian observational approach (Wendisch et al., 2024). The essence of this type of aircraft observations is illustrated in Fig. 3. Dropsondes launched from HALO and the airborne cloud radar sample the properties of an air parcel, e.g., air temperature  $T_1 = T(t_1, z_1)$ , air specific humidity  $q_1 = q(t_1, z_1)$ , and radar reflectivity  $Ze_1 = Ze(t_1, z_1)$  at a certain time  $t_1$  and at a specific altitude above ground  $t_1$  (alternatively at pressure altitude  $t_1$ ). These data are not collected for one air parcel at one altitude only, but for a column of vertically stacked air parcels as a function of altitude  $t_1$ . During HALO- $t_1$ , flight planning was based on trajectories calculated from forecasts available at that time. The forward trajectories originated from the stacked air parcels at the location of the first sampling at time  $t_1$ . In this way, flight patterns were designed to intercept as many of the air parcels observed in the stacked air column as possible at time  $t_1$  at a second time  $t_2$ .






To intercept the air parcels on their pathway, we performed forward-trajectory calculations using the hourly ICON simulations (Section 2.1) for 60 hours using the Lagrangian analysis tool LAGRANTO (Sprenger and Wernli, 2015). The height resolution of the starting points of the forward-trajectories was 5 hPa, resulting in an air column of 150 vertically stacked air parcels located between the surface (about 1000 hPa) and the top of the column corresponding to the average flight altitude of HALO (about 10 km, corresponding to roughly 250 hPa).

For each of the vertically stacked 150 air parcels observed at  $t_1$ , 30 regularly in latitude-longitude direction spaced trajectories were initiated within a radius of  $r=30\,\mathrm{km}$ , providing 4500 forward-trajectories vertically distributed over the entire column. If one of these 4500 trajectories initiated at  $t_1$  intersects with the vertical column sampled by HALO on its flight path at time  $t_2$  within a radius of 30 km, then we call it a matching trajectory. The vertically resolved dropsonde and HALO remote sensing measurements collected at  $t_2$  provide observations of, e.g.,  $T_2$ ,  $q_2$ , and  $Ze_2$ , which are then used in our analysis to quantify the changes of the thermodynamic and cloud properties of this same air parcel on its pathway (trajectory) by the difference between the observations collected at time  $t_1$  and  $t_2$ . A trajectory point at  $t_2$  is not necessary at the same altitude as it was at  $t_1$  (due to possible vertical movements of air parcels along their trajectories), and not all airmasses observed at  $t_1$  will also be observed at  $t_2$  (due to wind shear).

The procedure is repeated along the entire track of each HALO flight by initializing 4500 trajectories for each vertical column with a temporal resolution of 1 min. During HALO– $(\mathcal{AC})^3$ , the approximate flight time was about 8–10 hours, which means that more than  $4500 \, \mathrm{min}^{-1} \times 8 \, \mathrm{hours} \times 60 \, \mathrm{min}$  per hour=  $2.2 \times 10^6 \, \mathrm{air}$  parcel trajectories have been calculated for each HALO flight (Wendisch et al., 2024). More details on the assessment of the quality of the calculated forward-trajectories, the statistics and the vertical distribution of the relative number of matching trajectories (hit rate), and the vertical displacement of the air parcels along their trajectories are given in Appendix A.

To specifically address Objective 2, modeled tendencies along the trajectories were extracted from the hourly ICON output, always taking the closest time step. To meet Objective 3, the measurements and simulation results at starting point 1  $(T_1, q_1, q_2, \dots, q_n)$ 

and  $Ze_1$ ) and the matching point 2  $(T_2, q_2, \text{ and } Ze_2)$  are extracted. Subsequently, the temporal change rates of quantity  $\psi$  (with  $\psi$  representing T,  $\theta_e$ , q, RH, or Ze) were calculated:

$$\frac{\Delta \psi}{\Delta t} = \frac{\psi_2 - \psi_1}{t_2 - t_1} \,. \tag{1}$$

If  $\psi = T$  or  $\psi = \theta_e$  we call it the temperature change rate; if  $\psi = q$  or  $\psi = RH$  we use the term of humidity change rate.

# 3 Eulerian model evaluation: Comparison with dropsonde and radar measurements


In this section, we focus on the Objective 1 of this paper. We compare the results of the ICON simulations with the observations acquired during the HALO flights within an Eulerian framework. Specifically, we investigate the ability of the ICON model to reproduce the vertical profiles of thermodynamic measurements from dropsondes  $(T, \theta_e, q, RH)$ , and the cloud data (Ze) over open ocean and sea ice.

Figure 3. Sketch of quasi-Lagrangian flight strategy. The air parcels are indicated by white ellipses. They are formed by a vertical extension of  $\Delta p = 5$  hPa and a horizontal circular horizontal area with radius r = 30 km. Thus, the evolution of thermodynamic variables (e.g., temperature T, specific humidity q, radar reflectivity Ze) over time t and the involved processes can be studied along trajectories (dashed lines). As an example, a matching trajectory is indicated by a thick dashed gray line. This figure represents a modified version of Fig. 2 by Wendisch et al. (2024).

# 3.1 Thermodynamic variables – Case studies







Figure 4 presents vertical profile measurements collected with dropsonde of air temperature  $(T_{\rm meas})$  and specific humidity  $(q_{\rm meas})$  as a function of altitude above ground (z) for the two case studies of a WAI and a CAO. In addition, the corresponding ICON simulations  $(T_{\rm ICON})$  and  $(T_{\rm ICON})$  and the difference between the ICON simulations and the dropsonde measurements are shown. The profiles of measured and modeled equivalent potential temperature  $(\theta_{\rm e})$  and relative humidity (RH), instead of T and  $T_{\rm eq}$  are provided in Appendix B (Fig. B1).

During the WAI case, the lower parts of the airmass started with temperatures reaching values up to about 6–7 °C over the open ocean surface, far away from the Marginal sea Ice Zone (MIZ) (Fig. 4a and 4b, lower panels). When the warm airmass moved northward, then reaching the MIZ (yellow lines), and subsequently arriving over the sea ice (blueish lines), the near-surface air temperatures gradually decreased, but did not match the sea ice surface (skin) temperature of no greater than 0 °C. This result may be interpreted as an indication of the fact that the cooling through turbulent heat fluxes of the near-surface airmass on its way to the north lags slightly behind the actual sea ice skin temperature. However, it should be kept in mind that in this specific measurement flight, the dropsondes launched over the sea ice sampled the airmass close to the MIZ (Fig. 2a), thus giving the airmass only little time to adjust to the cold sea ice surface. This also explains the low variability of the soundings over sea ice, as compared to the larger spatial variability of the temperature profiles over open ocean, as they were made at different locations horizontally relative to the advecting airmass. Figure 4a also shows that the altitude of the measured near-surface air temperature inversion steadily increased from about 0.1 km over the open ocean to almost 0.4 km over the sea ice.

The two panels of Fig. 4c quantify the difference between the ICON-simulated and the dropsonde-measured air temperatures. For this comparison, the model column closest to the location of the dropsonde at the surface has been used. As the dropsonde is traveling in space, while the model column is constant, this introduces some uncertainty, especially in highly variable situations, such as the MIZ. On average, the values of this difference appear to be in the range of about  $\pm 1$  K, with slightly less deviations over sea ice. Some larger values of the ICON-measurement difference below 0.6 km altitude imply that ICON does not realistically reproduce the near-surface air temperature inversion. If the temperature inversion height is not matched by the simulations, larger deviations between measured and simulated temperatures are likely. Above 1 km altitude, the temperature difference appears to be slightly smaller over sea ice compared to the difference over open ocean. Below about 1 km altitude, there seems to be a cold bias of the ICON results (lower panel of Fig. 4c).

Figures 4d, 4e, and 4f present the corresponding results concerning the specific humidity (q) for the WAI case. Not surprisingly,  $q_{\text{meas}}$  is more variable over open ocean (red lines in upper panel of Fig. 4d) than over sea ice (blue) because the horizontal spread of observations is greater over the ocean (Fig. 2). Additionally, the difference between measured and modeled specific humidity is generally larger over open ocean than over the sea ice, although the specific humidity is quite small in this case. No general and consistent specific humidity bias of the ICON results is seen, except that the near-surface ICON-simulated specific humidity over sea ice is slightly too dry, in addition to the air temperature being modeled too cold. For the specific humidity,

similar to air temperature, it is concluded that the ICON simulations perform somewhat better over sea ice than over open ocean for this WAI case.




Figures 4g to 4l depict corresponding graphs for the CAO case observed on 01 April 2022. The spatial evolution of the ABL below 1.5 km altitude is apparent, with a heating, moistening, and deepening ABL as the airmass flows from the sea ice over the open ocean. From these graphs, a cold bias of the ICON temperature simulations below 0.4 km altitude of up to -4 K becomes obvious. This bias may be related to the fact that the measured altitude of the near-surface air temperature inversion is not well represented by ICON, in particular over sea ice and the MIZ. This becomes apparent by the jump of the values of the difference between the ICON-simulated and dropsonde-measured temperature from about -4 K (cold bias of ICON) at about 0.2 km altitude to positive values (2 K, warm bias) close to 0.4 km altitude indicated by the blueish and yellow lines in Fig. 4f (lower panel). Another interesting feature shows up by comparing the measured near-surface air temperatures with the surface skin observations indicated by full, colored dots in Fig. 4g (lower panel). Similar to the WAI discussed above, but much more obvious here, the southward moving cold and dry airmass takes time to adjust to the warmer ocean surface skin temperature. In this CAO case, the near-surface air is still at least 5 K colder than the surface skin temperature even after about 5 hours of advection south of the ice edge. The lowest altitude of the dropsonde measurements that characterized the near-surface air was typically between 3–15 m above ground, with most values around 5 m. This range results from the fact that temperature and humidity dropsonde data were recorded with a 2 Hz frequency, and the descent rate of the dropsondes was around 11 m s<sup>-1</sup>. Thus, a vertical resolution of the dropsonde measurements of about 5 m has been achieved (Vaisala, 2020; Ehrlich et al., 2025). With respect to specific humidity, the ICON simulations are very close to the measurements throughout the entire vertical profile, which is hardly surprising given the generally low values of specific humidity during this CAO event.

Figure 4. Comparison of vertical profiles of dropsonde-measured and ICON-simulated air temperature T, specific air humidity q, and their differences. The results for the case of 13 March 2022 (WAI) are shown in panels (a) to (f); those obtained for 01 April 2022 (CAO) are depicted in (g) to (l). Panels (a) and (g) show vertical profiles of the measured air temperature  $T_{\rm meas}$ ; (b) and (h) the model (ICON) results  $T_{\rm ICON}$ ; (c) and (i) ICON minus measured difference ( $T_{\rm ICON} - T_{\rm meas}$ ). Panels (d) and (j) depict the vertical profiles of measured specific humidity  $q_{\rm meas}$ ; (e) and (k) the vertical profiles of modeled specific humidity  $q_{\rm ICON}$ ; (f) and (l) the difference,  $q_{\rm ICON} - q_{\rm meas}$ . Panels (g) and (h) (lower parts) include surface skin temperature measurements (full dots). Similar to Fig. 2, the color of the lines of the lower panel (0–1 km altitude) indicate the temporal distance (in hours) the air parcel travels between the location where the sonde was launched to the Marginal sea Ice Zone (MIZ). If the temporal distance is negative then the air parcel is moving towards, if it is positive the air parcel moves away from the MIZ.

# 3.2 Clouds and precipitation using radar reflectivity – Case studies






To characterize cloud properties, we use vertical profile measurements of radar reflectivity (Ze) as a function of altitude (z) as a proxy. In particular, we compare radar reflectivity measured along the HALO flight paths (Ze<sub>meas</sub>) with corresponding simulations by the PAMTRA algorithm (Mech et al., 2020) based on ICON output (Ze<sub>ICON</sub>) including all hydrometer classes. The large number of measured and simulated profiles allows for a statistical evaluation using joint histograms of altitude and reflectivity, so-called contoured frequency by altitude diagrams (CFADs). For both case studies (WAI on 13 March, and CAO on 01 April 2022), Figure 5 provides the CFADs for measurements and simulations separately over open ocean and sea ice. Note, that here the absolute number of counts (samples) in an altitude-Ze bin is plotted in color. Because we have the same number of measurements and simulations, we can directly subtract the numbers in each bin to create a difference CFAD (ICON/PAMTRA minus radar).

For the WAI, both measurements and simulations reveal the highest number of clouds above 6 km altitude with reflectivities below  $-20 \, \mathrm{dBZ}$ , which is typical for ice clouds. Looking at the differences, a narrower Ze distribution within the simulated compared to the measured radar reflectivity CFAD is revealed. This is a typical model feature, as the assumptions in the one-moment scheme cause a tight relation between hydrometeor mixing ratio and Ze and thus can not represent the full natural variability. Jacob et al. (2020) could demonstrate that the use of a two-moment scheme significantly increases the variability in the simulated Ze (their Fig. 2).

Over the open ocean, the CFADs of measurements and simulations show a relatively similar behavior, with Ze increasing towards the ground but being mostly below  $0\,\mathrm{dBZ}$ , which can be regarded as a rough threshold for precipitation. There is a slight underestimation in the occurrence of Ze larger than  $-20\,\mathrm{dBZ}$  (blue colors) and a more pronounced overestimation around the lowest Ze values (less than  $-35\,\mathrm{dBZ}$ ; red colors). The latter could be explained by a lower sensitivity than the nominal  $-40\,\mathrm{dBZ}$ . Interestingly, more clouds occur over sea ice, especially at high altitudes. The simulated cloud systems seem to reach only up to  $9\,\mathrm{km}$  altitude compared to  $10\,\mathrm{km}$  obvious in the measurements. The narrow Ze-distribution in the simulations is evident at all altitudes, resulting in a clear maximum of precipitation around  $10\,\mathrm{dBZ}$  close to the surface, while the distribution is more spread out in the observations, also reaching higher values up to  $25\,\mathrm{dBZ}$ . While these correspond to relatively low rain rates below  $1\,\mathrm{mm/h}$ , these are still remarkable given the high latitude.

For the CAO, hydrometer occurrence is mainly limited to low levels over the ocean, as convection becomes only active over the relatively warm open ocean surface. Over the sea ice, only shallow non-precipitating clouds occur with cloud top altitudes limited to below 1 km. These features are well reproduced by the ICON simulations. However, a lack of larger values of simulated reflectivities above 5 dBZ in the lowest kilometer is evident, which is compensated by too many reflectivities with values around 0 dBZ. In contrast to the WAI, where precipitation occurred in the form of liquid rain, the radar measurements of the CAO case features snowfall. Thus, the ICON bias might be either due to not capturing the snowfall or be caused by the model assumptions about the shape and size of the ice crystals. The latter might be likely as in situ measurements by the low flying Polar aircraft reveal frequent occurrence of riming affecting particle shape in a complex fashion (Schirmacher et al., 2024).

In summary, the simulations reproduce the main features of the two, rather different cases well. Some deviations occur that can be explained by the need of the microphysical scheme to simplify the complexity of hydrometeors.

Figure 5. Comparison of the count distribution of measured radar reflectivity,  $Ze_{\rm meas}$ ; simulated radar reflectivity based on PAMTRA driven by ICON output,  $Ze_{\rm ICON}$ ; as well as their difference as ICON minus measurement counts. The data is presented as a function of altitude z over the open ocean, panels (a)-(c) and (g)-(i), and over sea ice, panels (d)-(f) and (j)-(l). The results for the WAI case observed on 13 March 2022 are presented in panels (a) to (f), and those obtained for the CAO case sampled on the 01 April 2022 are depicted in panels (g) to (l). Panels (a), (d), (g), and (j) show the vertical profiles of radar-measured reflectivity  $Ze_{\rm meas}$ , panels (b), (e), (h), and (k) depict the simulated radar reflectivity  $Ze_{\rm ICON}$ , and panels (c), (f), (i), and (l) show the respective radar reflectivity count difference  $\Delta$ Counts.

# 3.3 Evaluation of the entire data set of 12 flights





Table 1 quantifies the measurement-model comparisons in terms of Mean Absolute Error (MAE) and bias, averaged over the vertical profile data for altitudes below 1 km. These results are based on, and quantitatively complement the data of Subsections 3.1 and 3.2.

The ICON simulations of air temperature below 1 km altitude are generally quite accurate. For the two case studies, the calculated MAE values over sea ice range between 0.7 K (WAI on 13 March 2022) and 1.0 K (CAO on 01 April 2022). Corresponding MAE values over sea ice obtained for the entire data set are only slightly larger (1.1 K to 1.3 K for all 12 cases). Over open ocean, the MAE values are even smaller (0.5 K to 0.7 K for the entire data set), thus the height-averaged accuracy of ICON temperature simulations below 1 km altitude appears to be systematically better over open ocean compared to over sea ice.

A general, systematic but slight cold bias between -0.5 K and -0.9 K of the ICON results is indicated for all investigated CAO cases over both sea ice and open ocean. This cold bias is less or not existing for WAIs with bias values up to -0.1 K. Thus, both the MAE and the cold bias values for altitudes below 1 km appear systematically larger for CAOs than for WAIs. It should be noted that in previous studies, numerical weather prediction and reanalysis products have typically reported a warm bias over Arctic sea ice. This has been attributed to the missing insulating snow layer over the sea ice (Batrak and Müller, 2019), but also to an overabundance of mixed-phase clouds causing exaggerated downward turbulent mixing of atmospheric heat (Tjernström et al., 2021). In addition, the warm bias is often related to too large roughness lengths and exchange coefficients applied in parameterization of turbulent surface fluxes under stable stratification (Cuxart et al., 2006). Also, overabundance of clouds causes excessive thermal-infrared heating of the snow/ice surface (Tjernström et al., 2008), which is reflected as a warm bias in near-surface air temperature.

Not surprisingly, similar conclusions with regard to MAE and bias can be drawn for the equivalent potential temperature: the height-averaged accuracy of ICON simulations below 1 km altitude appears better over open ocean compared to over sea ice, and a cold bias of ICON simulations as compared to the measurements is on average larger for CAOs than for WAIs.

Specific humidity and relative humidity are well reproduced by the ICON simulations. For specific humidity, the MAE and bias are on average smaller for CAOs than for WAIs, and they are mostly smaller over sea ice compared to over open ocean for both types of conditions. Since relative humidity also depends on temperature, the comparison statistics for relative humidity do not have any consistent patterns. Overall, MAE values of relative humidity for all sub-categories are less than 10%.

For the radar reflectivity, MAE is larger for the WAI compared to CAO, in part because of far more cloud observations in the WAI considered here. Apart from over sea ice in the WAI, the mean biases are negative (ICON simulating clouds and precipitation that are too weak). However, the standard deviation is much larger than the mean bias for all conditions, indicating that in spite of the mean biases there are plenty of individual observations with both positive and negative biases.

**Table 1.** Evaluation of ICON versus measurement results for the case study of a WAI observed on 13 March 2022, the case study on 01 April 2022 (CAO), and aggregated results of six WAIs and six CAOs observed during HALO– $(AC)^3$ . Given are the Mean Absolute Error (MAE) and the bias of ICON results, calculated for the lowest 1 km above ground. MAE is calculated as the vertical average of the absolute differences between ICON results and measurements with dropsondes (air temperature, T, equivalent potential temperature,  $\theta_e$ , specific air humidity, T, relative air humidity, T, and radar (radar reflectivity, T).

| Variable       | Unit           | Surface    | 13 March        | 1 2022 WAI       | 01 April 2022 CAO |                  |  |
|----------------|----------------|------------|-----------------|------------------|-------------------|------------------|--|
|                |                |            | 20 Dro          | opsondes         | 41 Dropsondes     |                  |  |
|                |                |            | MAE             | bias             | MAE               | bias             |  |
| T              | K              | sea ice    | $0.7 \pm 0.3$   | $-0.3 \pm 0.6$   | $1.0 \pm 0.2$     | $-0.8 \pm 0.2$   |  |
|                |                | open ocean | $0.6 \pm 0.2$   | $-0.2\pm0.4$     | $0.8 \pm 0.3$     | $-0.7 \pm 0.5$   |  |
| $	heta_{ m e}$ | K              | sea ice    | $1.7 \pm 0.6$   | $-1.4\pm0.6$     | $1.1\pm0.2$       | $-1.0\pm0.3$     |  |
|                |                | open ocean | $0.6 \pm 0.2$   | $0.1\pm0.5$      | $1.0 \pm 0.4$     | $-0.9 \pm 0.6$   |  |
| q              | $\rm gkg^{-1}$ | sea ice    | $0.18 \pm 0.07$ | $-0.09 \pm 0.16$ | $0.06 \pm 0.02$   | $-0.04 \pm 0.03$ |  |
|                |                | open ocean | $0.39 \pm 0.28$ | $-0.18 \pm 0.40$ | $0.08 \pm 0.04$   | $-0.05 \pm 0.06$ |  |
| RH             | %              | sea ice    | 1 ± 1           | $-1\pm1$         | $5\pm 2$          | $-2\pm3$         |  |
|                |                | open ocean | 8 ± 6           | $-3\pm8$         | $7\pm3$           | $-1\pm4$         |  |
| Ze             | dBZ            | sea ice    | $18 \pm 15$     | $4\pm23$         | $7\pm17$          | $-6\pm17$        |  |
|                |                | open ocean | $14 \pm 24$     | $-10 \pm 26$     | $9 \pm 15$        | $-5 \pm 16$      |  |
| Variable       | Unit           | Surface    | All Si          | ix WAIs          | All Six CAOs      |                  |  |
|                |                |            | 114 Dr          | opsondes         | 133 Dropsondes    |                  |  |
|                |                |            | MAE             | bias             | MAE               | bias             |  |
| T              | K              | sea ice    | $1.1 \pm 0.4$   | $0.0\pm0.6$      | $1.3 \pm 0.3$     | $-0.9 \pm 0.3$   |  |
|                |                | open ocean | $0.5 \pm 0.3$   | $-0.1 \pm 0.4$   | $0.7 \pm 0.4$     | $-0.5 \pm 0.5$   |  |
| $	heta_{ m e}$ | K              | sea ice    | $1.4 \pm 0.5$   | $-0.1\pm0.9$     | $1.4 \pm 0.3$     | $-1.0\pm0.4$     |  |
|                |                | open ocean | $0.9 \pm 0.5$   | $-0.3 \pm 0.9$   | $0.9 \pm 0.5$     | $-0.7 \pm 0.8$   |  |
| q              | $\rm gkg^{-1}$ | sea ice    | $0.22 \pm 0.10$ | $-0.04 \pm 0.16$ | $0.10 \pm 0.02$   | $-0.03 \pm 0.05$ |  |
|                |                | open ocean | $0.27 \pm 0.16$ | $-0.01 \pm 0.27$ | $0.17 \pm 0.08$   | $-0.06 \pm 0.15$ |  |
| RH             | %              | sea ice    | 8 ± 4           | $-1\pm6$         | $8\pm 2$          | $-1 \pm 4$       |  |
|                |                | open ocean | $6 \pm 4$       | $1\pm 6$         | $9 \pm 4$         | $-1\pm7$         |  |

# 4 Modeling of airmass transformations along matching trajectories

Building upon the overall good performance of the ICON model demonstrated in Section 3, we proceed with Objective 2 of this paper and investigate in this Section 4 the airmass transformations along the matching trajectories as they evolve during the two cases of WAI and CAO. Furthermore, we discuss the impact of processes driving these airmass changes. For this purpose, time series of change/process rates of air temperature (heating and cooling) and humidity (drying and moistening) are derived from corresponding hourly ICON forecasts. These thermodynamic change/process rates are plotted along the matching trajectories derived from the LAGRANTO tool as a function of the advective, temporal distance of the air parcel from the MIZ to illustrate the influence of surface types (sea ice, open ocean). We call this type of figures the "macaroni plots".

At the beginning of this section, time series of cloud and precipitation liquid water and ice contents are plotted along the matching airmass trajectories to evaluate phase transitions during the two WAI and CAO cases. Then, the importance of adiabatic and diabatic processes in general, and specifically the impact of selected diabatic processes (i.e., radiative, latent, turbulent) on the temperature change/process rates are quantified. At the end of this section, the drying and moistening of the air parcels along the matching trajectories are investigated by looking at the corresponding humidity change rates.

To enhance the clarity of the "macaroni plots", we consider in Section 4 only a subset of 1200 matching trajectories.

#### 4.1 Phase changes during cloud and precipitation evolution



The evolution of cloud phases for both case studies (13 March 2022, WAI; 01 April 2022, CAO) is shown in Fig. 6. For the WAI, a significant amount of liquid water evolves, starting somewhat before the MIZ, but enhancing significantly near the ice edge and somewhat over the sea ice as the trajectories lift (Fig. 6a). Cloud ice, snow, and graupel develop, mostly after the air mass has traveled over the sea ice for about 4 hours (Fig. 6c). Most solid phase comes in the form of graupel and snow, which forms over a deep layer most intensively below about 3 km once the airmass moves far enough over the sea ice (Fig. 6c). Interestingly, much of this precipitation appears to come at the expense of the liquid water with a significant transition at about 4 hours of advection time from the ice edge. For the CAO, the cloud phase evolution is straightforward. Liquid water forms at the top of the lifting cloud as the airmass moves over the open ocean (Fig. 6b). From this liquid cloud, ice, snow, and graupel forms and falls down towards the surface (Fig. 6d).

**Figure 6.** Subset of 1200 matching trajectories indicating the altitude of air parcels as a function of the temporal distance to the MIZ. The subset is chosen such that the plots are well covered and not overcrowded. The color corresponds to the liquid water and ice contents of clouds and precipitation simulated by ICON, respectively. The results for the WAI sampled on 13 March 2022, are shown in the left column in panels (a) and (c), those of 01 April 2022, when a CAO was observed, are depicted in the right column of panels (b) and (d). In all plots, airmasses move from left to right. Panels (a) and (b) show combined cloud liquid plus rain water contents, and panel (c) to (d) cloud ice as well as graupel and snow ice water contents.

#### 4.2 Heating and cooling of air parcels



# 4.2.1 Evaluation of adiabatic versus diabatic processes

The time series of total (adiabatic plus diabatic) temperature change rates (indicative of heating or cooling of the respective air parcel) are computed using the ICON output of air temperature that was saved during the ICON model runs with a one-hour temporal resolution. Specifically, these temperature change rates are estimated by the finite differences, described by Eq. 1, of temperature values that are one hour apart along the matching trajectories. The time-series of the 1-hourly temperature change rates are down-scaled to 1 minute temporal resolution by means of linear interpolation between the calculated hourly values.

The resulting total (adiabatic plus diabatic) temperature change rates are plotted in Figs. 7a and 7b for the two case studies of a WAI and a CAO considered in this paper.

To discriminate between adiabatic and diabatic effects, we calculate the temperature change rates caused by adiabatic processes (descent, ascent). For this purpose, the pressure changes along the matching trajectories are used. The resulting temperature change rates caused by adiabatic descent (heating) or ascent (cooling) are depicted in Figs. 7c and 7d. Finally, the temperature change rates induced by diabatic processes were derived as the residual, i.e., the total minus the adiabatic temperature tendencies (Figs. 7e and 7f).

In both WAI and CAO cases, there is a general structure of relatively more adiabatic heating upstream of the MIZ and relatively more adiabatic cooling downstream, with the WAI structure being somewhat clearer than that for the CAO (Figs. 7c and 7d). This structure is consistent with the direction of flow, with subsidence upstream effectively driving the flow and ascending air downstream. In both cases, the downstream ascent is related to the advected air interacting with the new local surface. Adiabatic processes generally dominate except for at the lowest levels over the downstream "target" area, where diabatic change rates can be significant (Figs. 7e and 7f). For WAI conditions buoyancy related to the warm air interacting with the cold surface drives the downstream ascent, while in CAO cases there is weaker ascent as the cold advecting air interacts with the spatially increasing boundary layer depth driven by strong surface turbulent heat fluxes.


Figure 7. The same "macaroni plots" as Fig. 6. However, the color corresponds to the temperature change rates simulated by ICON with blue depicting cooling and red representing heating of the air parcel. The results for the WAI (13 March 2022) are shown in the left column in panels (a), (c), and (e), those for the CAO (01 April 2022) are depicted in the right column by panels (b), (d), and (f). Panels (a) and (b) show the total (adiabatic plus diabatic) temperature change rates along the matching trajectories; panels (c) and (d) the adiabatic temperature change rates caused by descent and ascent of the air parcels, and panels (e) and (f) the diabatic portion of the total, temperature change rates derived as the residuum between total minus adiabatic temperature tendencies.

# 4.2.2 Importance of diabatic effects: Radiation, latent heat, and turbulence



To further explore the diabatic processes, we use the temperature process rates (in units of  $Kh^{-1}$ ) that are saved from the ICON output every hour during the forecast (Subsection 2.1). These rates represent results from parameterizations of temperature changes caused by radiative, microphysical, and turbulent processes. The parameterized temperature process rates are interpolated at the hourly positions to one-minute values and plotted along the matching trajectories (Fig. 8).

Figure 8a illustrates a weak radiative cooling throughout the entire column of the warm and humid airmass moving northward in the WAI case. This cooling is caused by emission of thermal-infrared radiation during its transport and changes based on variation in the atmospheric opacity. The CAO case reveals a distinct cloud top cooling and a near-surface heating as soon as the airmass reaches the warm open ocean (Fig. 8b). The radiative cooling is caused by emission of thermal-infrared radiation at cloud top (see also Fig. 6b). The radiative heating is due to absorption of thermal-infrared radiation below cloud base, which is emitted by the warm open ocean surface below and the cloud above.

Figures 8c and 8d show heating and cooling effects caused by latent heat release or consumption during phase transitions in clouds and precipitation, primarily over the downstream region of the trajectories for each case. Figure 8c shows that over the sea ice, the warm and humid airmass in this WAI experiences some latent heating due to mid-level snow and graupel formation (Fig. 6c). The results for the CAO presented in Fig. 8d indicate latent heating in the upper cloud parts due to condensation. Below cloud base, over the warm open ocean, cooling by latent heat consumption is caused by evaporation of precipitation.

Figures 8e and 8f illustrate the residual temperature process rates, which are mainly caused by turbulence. These are derived from:

- The temperature change rates caused by diabatic processes (Figs. 7e and 7f),
  - Minus the temperature effects due to terrestrial (thermal-infrared) radiative processes (Figs. 8a and 8b),
  - Minus the temperature impact caused by latent heat release or consumption (Figs. 8c and 8d), and
  - Minus minor contributions from subgrid-scale condensation, solar radiation, and convection (not shown).

Our use of the residual instead of the temperature process rates caused by total turbulence directly accessible from ICON is motivated by the following. The total turbulent temperature process rates computed and saved by ICON each hour include not only surface effects where energy is directly injected into or absorbed from the atmosphere, but also the turbulent mixing of neighboring airmasses that are, in particular, connected with the presence of clouds making the field of turbulence tendencies highly discontinuous both in space and time. However, the mixing of neighboring airmasses does not result in net (diabatic) energy changes of atmospheric layers. Using the residual temperature process rates, we thus mainly restrict the point of view to near-surface impacts. For the WAI case, the resulting Fig. 8e indicates strong cooling of near-surface air parcels over the cold sea ice due to turbulent processes, while aloft the pattern of turbulent heating is quite variable. For the CAO case, strong near-surface heating by turbulent processes is indicated over the warm open ocean (Fig. 8f), while weak cooling occurs in the cloud layer, counteracting some of the latent heat released there.

**Figure 8.** The same "macaroni-plots" as shown in Fig. 7, but here the effects of diabatic processes (radiative, latent, and turbulent) on temperature process rates are illustrated. Shown are the diabatic process rates determining heating and cooling of air parcels related to terrestrial (thermal-infrared) radiative energy fluxes in panels (a) and (b), latent heating and cooling in panels (c) and (d), and turbulent energy processes in panels (e) and (f).

#### 4.3 Drying and moistening of air parcels along matching trajectories

- Here we make use of the hourly specific and relative humidity output from the ICON model. Following the general procedure given by Eq. 1, we calculate, in one-hour time steps along the matching trajectories, the running average of the hourly values of specific or relative humidity provided by ICON and divide it by one hour. These values are interpreted as humidity change rates (in units of g kg<sup>-1</sup> h<sup>-1</sup>). These tendencies with hourly resolution are interpolated to one-minute values and plotted along the matching trajectories in color code (Fig. 9).
- Figure 9a shows a general decreasing tendency of specific humidity in the WAI airmass within most of the clouds (0–4 km). However, relative humidity is variable in large part (Fig. 9c) because of the significant variability in heating and cooling via turbulent processes (Fig. 8e) and the spatially variable formation and evaporation of condensed cloud mass. For the CAO case, it is interesting that there is a general increase of specific humidity over the growing ABL with little change above (Fig. 9b). However, from a relative humidity perspective (Fig. 9d) there is an increase where there is net diabatic cooling (radiative plus turbulent), which helps to drive condensation, and a decrease where there is net diabatic heating, contributing to the evaporation of precipitation in that region.

**Figure 9.** The same "macaroni plots" as shown in Fig. 7, but here the color corresponds to the humidity (specific and relative humidity) change rates simulated by ICON with blue depicting drying and red representing moistening of the air parcel. The results for the WAI sampled on 13 March 2022, are shown in the left column in panels (a) and (c), those of 01 April 2022, when a CAO was observed, are depicted in the right column of panels (b) and (d). Panels (a) and (b) show specific humidity tendencies, and panel (c) and (d) relative humidity tendencies.

#### 5 Quasi-Lagrangian model evaluation: Comparison of change rates

430

In the next step we pursue Objective 3 of this paper by investigating and comparing measured and modeled vertical profiles of change rates of the thermodynamic properties, which quantify the airmass transformations of air parcels transported in WAIs and CAOs. Specifically, the change rates,  $\Delta\psi/\Delta t$ , with  $\psi$  representing  $T, \theta_{\rm e}, q$ , or RH, are derived from the difference between the value of  $\psi$  obtained at the end  $(t_2)$  and start times  $(t_1)$  of each matching trajectory using Eq. 1. The change rates are inferred either from the measurements with dropsondes or from corresponding quantities calculated by ICON. Furthermore, we quantify the bias between ICON-derived, modeled and dropsonde-measured change rates.

Fig. 10 depicts the resulting change rates for T and q in the form of count distributions as a function of altitude for the two specific cases of WAI (13 March 2022) and CAO (01 April 2022). A corresponding plot for the change rates of  $\theta_e$  and RH is presented in Appendix C (Fig. C1). Figure 10a illustrates a distinct although small cooling of the airmass that is most obvious below about 3 km altitude, with largest values of  $-0.6 \, \mathrm{K} \, \mathrm{h}^{-1}$  close to the ground. ICON reasonably reproduces this cooling (Fig. 10b), although for altitudes less than 1 km the model yields systematically too little cooling compared to the measurements. Figures 10c and 10d show corresponding results for specific humidity. The northward moving humid airmass mostly dries by maximum values of up to  $-0.2 \, \mathrm{g} \, \mathrm{kg}^{-1} \, \mathrm{h}^{-1}$  in an altitude range between the surface and about 6 km. These values are similar to the drying rates estimated from an airborne moisture budget derived by Dorff et al. (2025) for the ARclassified WAI case on 15 March 2022. Actually, the drying of up to  $-0.2 \, \mathrm{g} \, \mathrm{kg}^{-1} \, \mathrm{h}^{-1}$  is mostly quite well represented by ICON, except in the lowest 1.5 km.

440

445

For the CAO case, Fig. 10e illustrates a significant heating of the airmass of up to 5 K h<sup>-1</sup>, which is mostly restricted to altitudes less than 1 km. Above 1 km altitude, the airmass does not adapt towards the higher temperature of the warm open ocean surface. As opposed to the low-level challenges for the WAI case, ICON reproduces this low-level warming in the CAO. Figures 10g and 10h show how humidity is picked up from the warm open ocean surface during the southward airmass transport and again ICON represents this moistening.

It is interesting to note that the structure of the temperature and humidity change rates found here closely resembles the results obtained from another CAO event, albeit for a substantially deeper boundary layer, and in a spatial, rather than time perspective (Kähnert et al. (2021), their Fig. 5). The correspondence between the quasi-Lagrangian results obtained here and the Eulerian results from Kähnert et al. (2021) probably reflect the quasi-stationary flow often found in CAOs, and point to the potential complementarity between time change rates diagnostics along trajectories and individual tendency output from model parameterizations.

Figure 10. Comparison of change rates derived from the quasi-Lagrangian measurements and simulated by ICON at the end and the start of the matching trajectories. Results obtained for the case on 13 March 2022 (WAI) are shown in the top panels (a) to (d); those for the case of 01 April 2022 (CAO) in the bottom panels (e) to (h). Panels (a) and (e) show observed change rates of air temperature  $\Delta T/\Delta t$ . Panels (c) and (g) depict the observed specific humidity change rates,  $\Delta q/\Delta t$ . Panels (b), (d), (f), and (h) illustrate respective differences (biases) of ICON simulation results minus the observations.

Looking at the entire data set of six WAIs and six CAOs, Table 2 shows for the WAI case, that the airmass cools and dries near the surface as it moves northward (see also lower parts in Figs. 10a and 10c), yet the relative humidity actually slightly increases, see Appendix C, Fig. C1 panel (c), indicating that the cooling effect on relative humidity is acting faster than the drying. In the ICON model, the cooling and drying appears to be slower than observed, and on balance the increase in relative humidity is also too slow. These general results also mean that the WAI case is generally representative of the full WAI data set.

For the CAO case, Table 2 indicates a heating and moistening of the layer below 1 km, which is consistent with the lower panels of Figs. 10e and 10g. The moistening effect outweighs the heating effect on relative humidity, such that the relative humidity also tends to increase. The rate of relative humidity increase is underestimated in the ICON simulations due to an overestimation of the heating rate and an underestimation of the moistening rate.

**Table 2.** Evaluation of ICON results of change rates for the 13 March 2022 (WAI), 01 April 2022 (CAO), and aggregated WAIs and CAOs of HALO– $(AC)^3$ . Given are the mean change rates as derived from observations, the mean absolute error (MAE) and bias of ICON. All data is calculated for the lowest 1 km above ground.

| Variable                                      | Unit                  | 13 1             | March 2022 W    | AI              | 01 April 2022 CAO |                 |                  |  |
|-----------------------------------------------|-----------------------|------------------|-----------------|-----------------|-------------------|-----------------|------------------|--|
|                                               |                       | 2                | O Dropsondes    |                 | 41 Dropsondes     |                 |                  |  |
|                                               |                       | obs. mean        | MAE             | bias            | obs. mean         | MAE             | bias             |  |
| $\frac{\Delta T}{\Delta t}$                   | ${\rm K}{\rm h}^{-1}$ | $-0.2 \pm 0.2$   | $0.2\pm0.1$     | $0.1\pm0.2$     | $1.5\pm1.4$       | $0.7\pm0.7$     | $0.1 \pm 1.0$    |  |
| $\frac{\Delta \theta_{ m e}}{\Delta t}$       | ${\rm K}{\rm h}^{-1}$ | $-0.3 \pm 0.2$   | $0.3\pm0.2$     | $0.2\pm0.3$     | $1.9\pm1.7$       | $0.8\pm0.8$     | $0.1\pm1.1$      |  |
| $rac{\Delta q}{\Delta t}$                    | $\rm gkg^{-1}h^{-1}$  | $-0.05 \pm 0.05$ | $0.05 \pm 0.03$ | $0.03 \pm 0.05$ | $0.13 \pm 0.11$   | $0.07 \pm 0.07$ | $-0.01 \pm 0.10$ |  |
| $\frac{\Delta RH}{\Delta t}$                  | $\%\mathrm{h}^{-1}$   | $0.3 \pm 1.5$    | $0.6\pm0.7$     | $-0.1 \pm 1.0$  | $2.5 \pm 10.4$    | $7.2 \pm 7.1$   | $-0.3 \pm 10.1$  |  |
| Variable                                      | Unit                  | All Six WAIs     |                 |                 | All Six CAOs      |                 |                  |  |
|                                               |                       | 114 Dropsondes   |                 |                 | 133 Dropsondes    |                 |                  |  |
|                                               |                       | obs. mean        | MAE             | bias            | obs. mean         | MAE             | bias             |  |
| $\frac{\Delta T}{\Delta t}$                   | ${\rm K}{\rm h}^{-1}$ | $-0.3 \pm 0.2$   | $0.2 \pm 0.1$   | $0.1\pm0.2$     | $1.1\pm1.4$       | $0.6 \pm 0.7$   | $0.1 \pm 0.9$    |  |
| $\frac{\Delta \theta_{\mathrm{e}}}{\Delta t}$ | ${\rm K}{\rm h}^{-1}$ | $-0.2 \pm 0.2$   | $0.3\pm0.2$     | $0.1\pm0.3$     | $1.6\pm1.5$       | $0.7\pm0.8$     | $0.1\pm1.1$      |  |
| $rac{\Delta q}{\Delta t}$                    | $g kg^{-1} h^{-1}$    | $-0.05 \pm 0.05$ | $0.05 \pm 0.04$ | $0.02 \pm 0.05$ | $0.09 \pm 0.12$   | $0.08 \pm 0.07$ | $-0.02 \pm 0.11$ |  |
| $\frac{\Delta RH}{\Delta t}$                  | $\%\mathrm{h}^{-1}$   | $0.5 \pm 1.4$    | $0.9 \pm 1.0$   | $0.0 \pm 1.2$   | $1.7 \pm 9.5$     | $7.0 \pm 6.7$   | $-1.2 \pm 9.5$   |  |

#### 465 6 Summary and conclusions



Comprehensive aircraft measurements and extensive numerical simulations were carried out to test how well the observed airmass properties and their transformations during WAIs and CAOs are captured by limited area simulations with the ICON (Icosahedral Nonhydrostatic) numerical weather prediction model. The observations were collected using the High Altitude and Long Range Research Aircraft (HALO) during a field campaign that took place in the European Arctic in March and April 2022 (Wendisch et al., 2024; Walbröl et al., 2024). HALO was equipped with a variety of in-situ and remote sensing instruments (Ehrlich et al., 2025). Here we analyze the data from numerous dropsondes launched during the HALO flights and measurements acquired by the cloud radar installed on HALO. Specifically, the observations used in this paper include vertical profiles of air temperature, humidity, and cloud properties. Six WAIs and six CAOs were sampled during the campaign and analyzed in this paper with two specific cases evaluated in detail: a WAI observed on March 13, 2022, and a CAO of April 1, 2022. The flight paths of HALO were carefully planned to allow both Eulerian and quasi-Lagrangian sampling. A purely Lagrangian measurement approach is not possible for aircraft measurements, as an aircraft generally flies much faster than the

slowly moving airmass. Therefore, we have introduced a sampling technique that attempts to observe the same air parcel at least twice on its flight path north during a WAI or on its way south within a CAO. This observation technique is termed the quasi-Lagrangian method. Such an approach requires careful flight planning with accurate trajectory simulations. During the campaign, we used trajectories based on the output of different numerical weather forecast models to plan the flight paths. For this work, we recalculated the trajectories using the wind fields provided by the ICON model.






As it turned out, the careful flight planning during the campaign paid off, as we were indeed successful with our quasi-Lagrangian observational technique. Numerous matching trajectories were identified that allowed the use of two consecutive observations of the same air parcel to estimate the changes of thermodynamic and cloud-related parameters along the trajectories. We have shown that during the six WAI cases analyzed here with rather complex wind fields, between 2% and 9% of the trajectories initialized along the HALO flight path actually hit the measurement volume of the HALO instruments (dropsonde and cloud radar) a second time. The proportion of these so-called matching trajectories was higher for less complex wind fields during CAOs (10% to 35%). The height-resolved analysis of the matching trajectories showed that the vertical distribution of the percentage of matching trajectories was quite homogeneous in most cases.

The observational and modeling results were compared in an Eulerian and quasi-Lagrangian framework. The Eulerian approach showed an overall good performance of the ICON results with differences between the modeled and measured temperatures of ±1 K averaged over the entire air column (0 km to 10 km). Below 1 km altitude, the mean absolute error (MAE) of the ICON-predicted air temperature compared to the measurements appeared smaller than 0.8 K over the open ocean; the corresponding MAE values over sea ice were smaller than 1.3 K. However, a systematic cold bias in ICON predictions of at most -0.9 K was observed, with largest magnitudes for CAOs. It also turned out that the altitude of the surface temperature inversion was not modeled accurately, mostly for CAOs over sea ice. It was also shown that the airmasses needed some time to adjust to the changing surface skin temperature; this time lag was obvious in both the dropsonde measurements and the ICON simulations. This adjustment occurs as a result of the turbulent heat fluxes between the surface and the lower atmospheric layers. This was most evident when cold airmasses moved from the sea ice over the warm open ocean during CAOs.

Specific humidity was well reproduced by the ICON model with MAE values averaged over the layer below 1 km altitude of less than 6.0 % (0.39 g kg<sup>-1</sup>), with largest values over the open ocean. A slight dry bias in specific humidity was observed in the ICON results with maximum values of 19.5 % (-0.18 g kg<sup>-1</sup>) derived over open ocean. MAE values for relative humidity were generally less than 10 % for the lowest 1 km. For cloud properties observed and modeled during WAIs, the radar reflectivity of the high- and low-level clouds and precipitation over the open ocean was underestimated in the simulations, but the radar reflectivity over sea ice was reasonably represented for most clouds. For CAOs, the radar reflectivity was underestimated at most altitudes.

The observations of change rates of thermodynamic properties showed that the warm and moist airmass of a specific WAI case cooled by about -0.3 to -0.5 K h<sup>-1</sup> on its way north at altitudes up to 8 km and dried by up to about -0.05 g kg<sup>-1</sup> h<sup>-1</sup> at a slightly lower altitude range. In a specific CAO case, the airmass warmed by up to 5 K h<sup>-1</sup> on its way south at altitudes of up to 1 km, and it picked up moisture of up to 0.4 g kg<sup>-1</sup> h<sup>-1</sup>. In both cases, these temperature and humidity variations were reproduced quite accurately by the simulations.

Additionally, it was shown that adiabatic processes dominated the heating and cooling of the air parcels over diabatic effects during WAIs and CAOs. Of the diabatic processes, latent heating and turbulent effects had a stronger impact on the temperature process rates of the air parcels than terrestrial radiative effects, especially over the warm ocean surface during CAOs.

Future aircraft campaigns should carefully consider the trade-offs between Eulerian and quasi-Lagrangian sampling strategies. While Eulerian sampling is broader and easier to implement, it lacks an inherent cause-effect relationship. In contrast, quasi-Lagrangian sampling is more constrained in space and time, but it directly captures airmass transformations along the large-scale flow. This distinction is critical, as Eulerian analyses may lead to misinterpretations about airmass evolution. Even for seemingly straightforward WAIs and CAOs, upstream conditions are not always directly linked to conditions much further downstream, which might be shaped by local effects and different environmental conditions. To mitigate biases in future campaigns, flight planning should ensure that trajectory times over open ocean and sea ice are comparable, reducing discrepancies in airmass history and transformations.

Collectively this analysis has demonstrated the great potential of the quasi-Lagrangian perspective. While there is some potential for true Lagrangian observations that follow advecting airmasses (Roberts et al., 2016), our quasi-Lagrangian approach provides a similar type of information that can be accomplished via carefully planned aircraft observations. We have demonstrated the ability to characterize airmass transformations by quantifying important parameters like the change of temperature and moisture in airmasses. Such analyses are essential to understand the life cycles of Arctic airmasses, how they evolve, and ultimately how they impact the other components of the Arctic system.

Data availability. The observational data used in this study is available from the PANGAEA Earth data repository: Flight tracks of HALO (Ehrlich et al., 2024), vertical thermodynamic and wind profiles from HALO dropsondes (George et al., 2024), radar reflectivities (Dorff et al., 2024), and skin temperatures (Schäfer et al., 2023). ERA5 is freely available on single levels, pressure levels, and model levels; for further information, refer to Hersbach et al. (2020). The ICON source code is freely available from GitLab (https://gitlab.dkrz.de/icon/icon-model/-/ tree/release-2024.01-public). Same-day trajectory matches during HALO-(AC)<sup>3</sup> based on ERA5 are also available from PANGAEA (Kirbus et al., 2024). Output from the ICON simulations, as well as all trajectory matches, are available from the authors upon request.

Author contributions. MW has designed and coordinated this study, and wrote the draft of the paper. BK conducted the trajectory analysis and plotted all figures. DO performed all simulations with ICON and wrote parts of the text. MDS, SC, and HS contributed to the data analysis and interpretation, as well as the language polishing of the paper. VS supported the ICON modeling and the discussion of the contents of the paper. All authors discussed the results and contributed to revisions of the text and the figures.

Competing interests. There are no competing interests to declare.



Acknowledgements. We gratefully acknowledge the funding by the Deutsche Forschungsgemeinschaft (DFG, German Research Foundation) - Project Number 268020496 - TRR 172, within the framework of the Transregional Collaborative Research Center "ArctiC Amplification: Climate Relevant Atmospheric and SurfaCe Processes, and Feedback Mechanisms  $(AC)^3$ ". We are grateful for funding of project grant number 316646266 by DFG within the framework of Priority Program SPP 1294 to promote research with HALO. The work by Davide Ori was funded by the DFG – Project number 443666877. This simulations performed within the framework of this paper also used resources of 545 the Deutsches Klimarechenzentrum (DKRZ) granted by its Scientific Steering Committee under project ID bb1086. The authors are grateful to AWI for providing and operating the two Polar 5 and Polar 6 aircraft. We thank the crews and the technicians of the three research aircraft for excellent technical and logistical support. The generous funding of the flight hours for the Polar 5 and Polar 6 aircraft by AWI. and for HALO by DFG, Max-Planck-Institut für Meteorologie (MPI-M), and Deutsches Zentrum für Luft- und Raumfahrt (DLR) is greatly appreciated. Matthew Shupe was supported by a Mercator fellowship with  $(\mathcal{AC})^3$  and by NOAA via NA22OAR4320151 and FundRef 550 https://doi.org/10.13039/100018302. We thank Jan Kretzschmar (Leipzig University), Andreas Walbröl (University of Cologne) and Heini Wernli (ETH Zürich) for helpful comments to the paper draft. This publication was supported by the Open Access Publishing Fund of Leipzig University.

# Appendix A: Trajectory assessment



The credible identification of matching trajectories is crucial for our study; it critically depends on the quality of the trajectory calculations, which were performed using LAGRANTO on the basis of ICON wind fields. To gain trust in the calculated trajectories, in a first approach, the results of the ICON simulations of the vertical profiles of the horizontal (zonal and meridional) wind speed components were compared with corresponding dropsonde measurements (Fig. A1). The wind fields determine the trajectories, thus their accuracy is important for reliable trajectory calculations. From Fig. A1 we find that the dropsonde data and the ICON simulations of the wind speeds agree in terms of Mean Absolute Error (MAE, 0–8 km altitude) of  $2.3\pm2.1$  m s<sup>-1</sup> with a bias of  $-0.3\pm3.1$  m s<sup>-1</sup> during the WAI observed on 13 March 2022. In the case of the CAO of 01 April 2022, the agreement is even better (MAE, 0–8 km altitude,  $1.3\pm1.4$  m s<sup>-1</sup> with a bias of  $0.03\pm1.9$  m s<sup>-1</sup>).

**Figure A1.** The same as Fig. 4 but for horizontal wind components U and V.

Secondly, we compare the trajectories from ICON with those derived from ERA5 (Hersbach et al., 2020). ERA5 wind data are available for 137 model levels, which are vertically spaced between the surface and the top of the atmosphere on a regular  $0.25^{\circ} \times 0.25^{\circ}$  latitude–longitude grid with a 1 hour temporal resolution. Trajectories were also calculated with LAGRANTO based on ERA5 wind fields, and matching trajectories were calculated in the same fashion as for ICON. For all flights we compared the absolute and relative numbers of matching trajectories (Figs. A2a and A2b). The absolute values of the numbers of matching trajectories are of the order of  $10^{5}$  for all flights except the 14 March WAI case, which demonstrates the statistical

significance of the trajectory dataset. The absolute numbers of matches are mostly smaller for the WAIs (12–20 March 2022) compared to the CAOs (21 March to 04 April 2022). However, the results using ERA5 and ICON wind fields to derive the trajectories by LAGRANTO agree well for all 12 flights. Panel (a) of Fig. A2 shows the relative fraction of trajectories that had matching observations. This fraction was obtained by dividing the absolute number of matching trajectories by the total number of initiated trajectories (roughly  $2.2 \times 10^6$ , depending on flight duration) for each flight. This figure effectively shows the hit rate of trajectories, quantifying the practical success of our quasi-Lagrangian observation strategy. For the WAI cases, the percentage fraction of matching trajectories is below about 10%, whereas for CAOs this percentage is mostly higher ranging between 5% and 35%. WAIs reach much higher vertically with embedded convection, causing more complicated wind patterns, which decrease the hit rate for matching trajectories. CAOs are most pronounced at lower altitudes with more uniform wind fields. This allows for more certain flight planning, which increases the hit rate of matching trajectories. Summarizing, Fig. A2, panel (a) reveals only minor differences when the LAGRANTO trajectories are calculated using wind fields provided by ERA5 versus ICON, which indicates consistency of the ERA5 and ICON wind data and additionally supports the reliability of the trajectory matching analysis.

Figure A2, panel (b) complements panel (a) by showing the relative (fraction) numbers of matching trajectories per flight using LAGRANTO (based on ICON 3D wind fields) as a function of pressure altitude of the start point of the trajectory at time  $t_1$ . The absolute number of matching trajectories for air parcels with a vertical extension of 25 hPa is of the order of up to  $10^4$  (not shown) giving sufficient statistical significance. The average relative fraction of the matching trajectories as a function of altitude shown in panel (b) is, similar to panel (a), in the range of mostly below 10 % for the WAI cases, and between 5–35 % for CAOs. For most flights, the vertical distribution of the percentage fractions of matching trajectories appears quite homogeneous.

**Figure A2.** Relative numbers of quasi-Lagrangian matches (matching trajectories, hit rates) for the research flights sampling WAIs (12–20 March 2022) and CAOs (21 March to 04 April 2022) during HALO– $(AC)^3$ . The trajectories were derived from ERA5 (blue) and ICON (black) wind fields. On each day, indicated on the abscissa axis, one HALO flight took place. Panel (a) includes relative (fractions) numbers of quasi-Lagrangian matches accumulated over each of the flights. Panel (b) plots the relative (fractions) numbers of quasi-Lagrangian matches in color code as a function of pressure altitude with a vertical resolution of 25 hPa. Vertical averaging of the colored columns of (b) corresponds to the values indicated by the vertical bars in panel (a).

Finally, we investigate the vertical displacement of the air parcels moving along trajectories by illustrating the matching trajectories for the two chosen case studies (13 March 2022, WAI, and 01 April 2022, CAO) in the form of a flight time – flight altitude plot in Fig. A3. This graphic depicts the altitude of the start points of matching trajectories  $z(t_1)$  at time  $t_1$  (orange dots) when the first sampling takes place (Fig. 3), and the altitude of the end the points of matching trajectories  $z(t_2)$  (red dots) where the second sampling occurred (at  $t_2$ ) during the HALO flight for the WAI (Fig. A3a) and CAO (Fig. A3b) cases. Some randomly selected examples of the height-dependent matching trajectories connecting start and end points are indicated by gray arrows. The arrows demonstrate that over the investigated time scale (i.e., within a single flight), the air parcels only slightly change altitude along the matching trajectories during the two cases investigated here.

590

Figure A3. Overview of start and end points of 1200 randomly selected matching trajectories during the flights conducted on (a) 13 March 2022, and (b) 01 April 2022. Orange dots denote the start point at altitude  $z(t_1)$  at the start time  $t_1$  of the matching trajectory (first sampling), and red dots indicate the altitude  $z(t_2)$  at the end time  $t_2$  of the matching trajectory where the second sampling occurred. Gray arrows show some randomly selected examples of the links between the start and end points of the matching trajectories. The sea ice concentration was extracted from ICON every minute at the respective position of HALO on that day and plotted at the bottom of the graph.

These results give high confidence in the reliability of the simulated forward-trajectories, which form the basis of the subsequent analysis of matching trajectories.

Appendix B: Eulerian comparison between ICON simulations and dropsonde measurements of equivalent potential temperature and relative humidity

**Figure B1.** The same as Fig. 4 but for equivalent potential temperature  $\theta_e$  and relative humidity RH.

# 600 Appendix C: Quasi-Lagrangian comparison between ICON simulations and dropsonde measurements of equivalent potential air temperature and relative humidity

Figure C1. The same as Fig. 10 but for the observed change rates of equivalent potential air temperature  $\Delta\theta_{\rm e}/\Delta t$ , and relative humidity change rates,  $\Delta RH/\Delta t$ .

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
