# Peer review of "Observed and modeled Arctic airmass transformations during warm air intrusions and cold air outbreaks"

_EGUsphere, 2025_

## Author Comment (AC1)

**Replies to the comments of three reviewers on our manuscript submitted to *Atmos. Chem. Phys.***

Wendisch, M., Kirbus, B., Ori, D., Shupe, M. D., Crewell, S., Sodemann, H., and Schemann, V.: Observed and modeled Arctic airmass transformations during warm air intrusions and cold air outbreaks, EGUsphere [preprint], https://doi.org/10.5194/egusphere-2025-2062, 2025.

We would like to sincerely thank the reviewers for their careful examination of our manuscript. We have carefully taken the reviewers' comments and suggestions into account, and our responses are given below.

In addition to the reviewers' comments, we have polished the text and figures without changing the scientific context. You can track the text changes in the "Diff" version provided.

The reviewer's comments are highlighted in bold below, our responses are in normal font, and the changes we have made to the text are shown in italics.

We truly appreciate your suggestions and the time you have spent reading this extensive manuscript.

With kind regards,

Manfred Wendisch, Benjamin Kirbus, Davide Ori, Matthew D. Shupe, Susanne Crewell, Harald Sodemann, and Vera Schemann

**The manuscript presents a detailed study on Arctic airmass transformations during on-ice and off-ice airflows. The study is based on extensive data set collected during the HALO-AC3 flight campaign mostly over the Barents Sea and Fram Strait. The research aircraft observations, including those made using tethersondes, are strongly supplemented by simulations using the high-resolution numerical weather prediction model ICON and by sophisticated trajectory calculations, including identification of matching trajectories of observations and simulations. According to my knowledge, the material gathered is more extensive than in any previous studies addressing Arctic airmass transformations. The analyses appear carefully made and yielded interesting result on the roles of adiabatic and diabatic processes, the latter including condensation/evaporation, radiative transport, and turbulent surface heat flux. In addition to improved process understanding, the study yielded new information on the performance of the ICON model. I suggest acceptance of the manuscript subject to minor revisions.**

Thank you very much for this overall positive evaluation.

**Detailed comments:**

**Introduction: It would be good to clearly summarize the relationship between this manuscript and the papers by Wendisch et al. (2023a, 2023b, 2024) cited in various parts of the manuscript. There seems to be a bit of overlap between them.**

This comment was also made by Reviewer #3.

The Wendisch et al. (2023a) paper is a general introduction into the $(\mathcal{AC})^3$ project and its first results, it mentions the HALO $-(\mathcal{AC})^3$ campaign but provides no detailed data analysis of the corresponding measurements. It should be noted that the $(\mathcal{AC})^3$ project involves much more than the HALO $-(\mathcal{AC})^3$ campaign, which is only one of the many activities within $(\mathcal{AC})^3$. The Wendisch et al. (2023b) paper considers one specific aspect of the influence of airmass transformations, namely the impact of surface properties on the radiative energy budget. That has not been investigated in the current manuscript. Furthermore, no data from the HALO $-(\mathcal{AC})^3$ campaign have been used in Wendisch et al. (2023b). The focus of the Wendisch et al. (2024) paper is the general introduction of the HALO $-(\mathcal{AC})^3$ campaign. In this paper the quasi-Lagrangian strategy is introduced, but only exemplarily measurements are shown. In the current manuscript we compare measured and ICON-simulated change rates of thermodynamic and cloud properties. In this regard, all cited papers cover different areas with only minimal overlapping in the description of the motivation and the methodology of the quasi-Lagrangian approach, which appear needed for stand-alone papers.

To clearly indicate the distinctions between the cited Wendisch et al. (2024) and the current submission, we have replaced the last two sentences of the second-last paragraph of the introductory section by:

*"HALO-$(\mathcal{AC})^3$ delivered numerous observations of thermodynamic and cloud properties along pronounced WAIs and CAOs over open ocean and sea ice, which have been introduced and summarized by Wendisch et al. (2024). This publication also motivated extensively the general need for a Lagrangian-based model evaluation and the required quasi-Lagrangian observations, including their practical realization by aircraft measurements. In the current study, we go one step beyond by exploiting the HALO-$(\mathcal{AC})^3$ measurements in synergy with simulations conducted with the ICON (Icosahedral Nonhydrostatic) weather forecast model to investigate airmass transformations during WAIs and CAOs."*

**Lines 17-18: Be more specific about the 50% decline of the Arctic sea ice cover: extent, thickness or volume? Annual mean of a certain season?**

Here we refer to the sea ice extent in September (annually averaged). Thanks for this hint; it was important to clarify this statement. We have changed the corresponding sentence to:

*"One of the most obvious signs of these changes is the almost 50 % decline of the Arctic sea ice extent detected in the time series of the monthly averaged September data since the 1970s (Stroeve et al. 2007, Olonscheck et al. 2019, Serreze et al. 2019, Screen et al. 2021), with a trend of –(11.8±1.3) % per decade for the years between 1979-2023 ([https://www.meereisportal.de/en/maps-graphics/sea-ice-trends#gallery-1](https://www.meereisportal.de/en/maps-graphics/sea-ice-trends#gallery-1))."*

**Line 63: Do you mean "In addition to these model difficulties …"?**

Yes, this suggestion makes total sense. We have replaced "In spite …" by "In addition …" and the revised sentence (now line 67) reads:

*"In addition of these model difficulties, there remains a general lack of observational data with which to evaluate the spatiotemporal evolution of cloudy airmass properties during synoptic-scale transport events, particularly near the surface."*

**Line 106-107: It is somewhat misleading to call energy and mass fluxes as surface properties. The fluxes may change in time even if such surface properties as ice concentration and thickness remain constant.**

Agreed. We have replaced the "surface properties" by "quantities". The revised sentence (now line 114) reads:

*"Also, quantities such as energy and mass fluxes are stored."*

**Lines 147-148: Is this a good argument to not consider the drift of drop sondes? A bit less than 30 km may matter quite a lot in the ice-edge zone, in particular during cold-air outbreaks.**

We always took the location of the dropsonde at its lowest point above ground as reference ($z \approx 0$ km). In this way the atmospheric boundary layer (with strong interactions with the underlying surface) is well accounted for. Furthermore, horizontal wind speeds were generally below 25 m s$^{-1}$ (Fig. A1). At a typical dropsonde descent rate of 11 m s$^{-1}$ (Vaisala 2020), a vertical drop of 1 km took the dropsonde around 90 seconds. This corresponds to a maximum horizontal drift of 2.3 km, or slightly less than the width of one grid cell (2.4 km). The lowest 2 km of the dropsonde descent thus corresponds to only two grid cells.

We have added the following text correspondingly:

*"Please note that the horizontal drift of the dropsondes during their vertical fall, which was always less than 30 km from release at HALO flight altitude to touchdown on the ground, was not taken into account. Considering the horizontal wind speeds, which were generally below 25 m s$^{-1}$ (Fig. A1), and the typical dropsonde descent rate of 11 m s$^{-1}$ (Vaisala, 2020), a vertical fall of 1 km takes the dropsonde around 90 seconds. This corresponds to a maximum horizontal drift of 2.3 km, which is slightly less than the width of one ICON model grid cell (2.4 km). If the dropsonde falls 2 km vertically, it drifts horizontally through only two grid cells, which should not significantly bias the Eulerian measurement-model comparison. Furthermore, the hourly model output was linearly interpolated to 1 min resolution, to match the temporal resolution of the PAMTRA simulations and to be much closer in time to the measurements."*

**Lines 226-227: One would expect that successful modelling of crossing of the ice edge could be a challenge. Hence, it is somewhat surprising that during a warm-air intrusion the ICON model performs somewhat better over sea ice than over the open ocean. On lines 283-284 you refer to systematically better ICON results over the open ocean than sea ice. Any ideas on this?**

Of course, there is always a difference between general behaviour / challenges – as getting the surface properties and in our case especially the input data reasonable over sea ice – and specific case studies as the warm air intrusion case. Furthermore, the influence of the sea ice and the challenge of representing the crossing of the ice edge will of course be more difficult and prominent during cold air outbreaks than during warm air intrusions, which are dominated by the warm and moist air and more large scale features. Nevertheless, to not lead to further confusion, we deleted the mentioned sentence in lines 226-227 (original manuscript), as the signal is also not too strong.

**Line 238: Specify the altitude, as the number (5 K) is probably very sensitive to it.**

Thanks for this hint, the text was not precise enough here. We have replaced "… than the surface …" by "… than the surface skin temperature …" in this sentence, which now reads:

*"In this CAO case, the near-surface air is still at least 5 K colder than the surface skin temperature even after about 5 hours of advection south of the ice edge."*

To characterize the lowest dropsonde measurement height, determining the near-surface air we have included the following text:

*"The lowest altitude of the dropsonde measurements that characterized the near-surface air was typically between 3-15 m above ground, with most values around 5 m. This range results from the fact that temperature and humidity dropsonde data were recorded with a 2 Hz frequency, and the descent rate of the dropsondes was around 11 m s$^{-1}$. Thus, a vertical resolution of the dropsonde measurements of about 5 m has been achieved (Vaisala 2020, Ehrlich et al. 2025)."*

**Lines 267-270: Could the bias be due to challenges in modelling the Lagrangian evolution of the airmass? By "too low reflectivities" (l278), do you mean errors in the PAMTRA algorithm?**

The bias could be caused by many different aspects. As mentioned in the text, the assumptions on shape and size of ice crystals and snow particles in the model might be too simplistic or well suited for this case – a special challenge during mixed-phase or ice cloud events. Another aspect might still be the model resolution – as clouds are not resolved at 2 km resolution an additional bias might be introduced by comparing area averaged / fractional cloud cover with remote sensing observations. This aspect will matter more during cold air outbreaks than the more large-scale driven warm air intrusions. All of the aspects could be and will be tackled by further studies – e.g. increasing the resolution and different sensitivity studies on the representation of ice crystals in the model. But this would require too much details for the current more general overview.

We slightly rephrased the sentence to be more precise:

*„Thus, the ICON bias might be either due to not capturing the snowfall or be caused by the model assumptions about the shape and size of the ice crystals."*

To answer the second part of the question of the reviewer: NO, PAMTRA is not the problem, and we sustain what is already written in the reply. Both resolution and particle properties are having an effect here. Perhaps one also needs to remember that those two effects are entangled, it is unlikely to me that massively rimed snowflakes are found everywhere in the 2.4 km area, hence also ICON is sort of correct while assuming average particle properties. Moreover, one needs to remember some lessons from statistics: While mass scales linearly with mass (trivial right?) reflectivity scales with mass-squared. ICON priority is to model

hydrometeor mixing ratio (and thus precipitation rate) therefore it is better to assume a particle that reflects the average mass of particles found in clouds, to model accurately reflectivity one needs to assume particles whit squared mass matching the average squared mass.

**Lines 288-291: Near-surface warm bias over Arctic sea ice is indeed common, and I fully agree on the two reasons mentioned in the text. In addition, the warm bias is often related to too large roughness lengths and exchange coefficients applied in parameterization of turbulent surface fluxes under stable stratification (e.g., Cuxart et al., 2006). Also, overabundance of clouds causes excessive longwave heating of the snow/ice surface (Tjernström et al.,2008), which is reflected as a warm bias in near-surface air temperature.**

Thanks, we have included this additional text and the two references into the manuscript, right after line 291.

*"In addition, the warm bias is often related to too large roughness lengths and exchange coefficients applied in parameterization of turbulent surface fluxes under stable stratification (Cuxart et al. 2006). Also, overabundance of clouds causes excessive thermal-infrared heating of the snow/ice surface (Tjernström et al. 2008), which is reflected as a warm bias in near-surface air temperature."*

**Line 313: humidity change rates**

Done. New text:

*"Lastly, the magnitude of humidity change rates are compared to the corresponding temperature change rates."*

**Line 315: What do you exactly mean by "such that the plots are well covered"?**

This sentence was deleted; it obviously causes confusion and does not really help. What we meant by it was that we did not want to overload the figures, which we said in the sentence before anyway.

**Line 321: Do you mean "moves far enough over the sea ice"?**

Yes, this is what we mean. We replaced the sentence by:

*"Most solid phase comes in the form of graupel and snow, which forms over a deep layer but most intensively below about 3 km once the airmass moves far enough over the sea ice."*

**Lines 341-342: Can you identify the reasons for descending flow upstream and ascending flow downstream in the cases of CAO and WAI?**

We have provided some more detail on the reasons for this flow structure by rearranging some of the content from the original paragraph and adding a few key points. The new paragraph states:

*"In both WAI and CAO cases there is a general structure of relatively more adiabatic heating upstream of the MIZ and relatively more adiabatic cooling downstream, with the WAI structure being somewhat clearer than that for the CAO (Figs 7c and 7d). This structure is consistent with the direction of flow, with subsidence upstream effectively driving the flow and ascending air downstream. In both cases, the downstream ascent is related to the advected air interacting with the new local surface. Adiabatic processes generally dominate except for at the lowest levels over the downstream "target" area, where diabatic change rates can be significant (Figs. 7e and 7f). For WAI conditions buoyancy related to the warm air interacting with the cold surface drives the downstream ascent, while in CAO cases there is weaker ascent as the cold advecting air interacts with the spatially increasing boundary layer depth driven by strong surface turbulent heat fluxes."*

**Figures 6-9: These are excellent figures with so many interesting and interpretable findings! In this respect, I am not sure if Figure 10 is the highlight of this paper (as stated on line 401).**

Well, Fig. 10 presents the essence of the paper by combining measurements and models. But you are right, maybe we should leave it to the reader to decide about what is a highlight of the paper. Therefore, we have deleted the beginning of the respective sentence (line 401) "As a highlight of this paper, …"

**Section 4.2.2: Referring to diabatic effects due to turbulence sounds vague. I suggest writing about convergence of surface sensible heat flux in the case of diabatic heating (divergence in the case of cooling).**

We are not sure that we understand the comment. The term "effect" is used sparsely in the section. We would agree that "processes" and "energy fluxes" are better terms with respect to "effects" sometimes, but also the concept of divergence of surface flux is mathematically maybe not correct. Moreover, we do not focus to the surface when referring to the effect of turbulence. We just accept the suggestion and possibly change the term effects a couple of times.

**Line 383: Fig. 9c**

We have not changed the figure number, but instead the sentence to:

*"Figure 9a shows a general decreasing tendency of specific humidity in the WAI airmass within most of the clouds (0-4 km)."*

**Line 385: Fig.9b**

We have changed the figure number and the text, the revised sentence reads:

*"For the CAO case, it is interesting that there is a general increase of specific humidity over the growing ABL with little change above (Fig.9b)."*

**Lines 415-416:  But in Figure 9d, near-surface relative humidity decreases downwind over sea ice. Any comment on this?**

The statement in Lines 415-416 was not fully correct and, therefore, has been removed from the manuscript.

**Line 425: This is interesting, as in many models the turbulent exchange coefficients for the turbulent surface fluxes of heat and moisture are identical. Is this the case also in ICON? Naturally the underestimation and overestimation may be related to later advection instead of vertical surface fluxes.**

ICON computes these fluxes in a complicated way. There is a combination of diffusion and turbulent transport, plus, all other surface processes (radiation, precipitation, condensation/evaporation, advection) act at the same time and the results in Table 2 estimate all of them. Surface processes are also treated in ICON by separate modules depending on the surface properties and it is not immediately clear how these modules interact with the turbulence solver. We cannot reliably answer the reviewer, but we also consider that knowledge not necessary to comment the presented results since, as written before, multiple processes (and not only turbulent surface fluxes) contribute to the temperature and moisture tendencies.

**Line 454: the error was smaller rather than better.**

Done:

*"Below 1 km altitude, the mean absolute error (MAE) of the ICON-predicted air temperature compared to the measurements smaller than 0.8 K over the open ocean; the corresponding MAE values over sea ice were smaller than 1.3 K."*

**Lines 451-474: To make it easier for a reader, I suggest dividing this long paragraph into two or three paragraphs, perhaps starting on lines 460 and 466.**

Agreed, and done!

**Lines 483-488: Considering true Lagrangian observations, the potential of controlled meteorological balloons deserves to be mentioned. There are papers by, e.g., Lars Hole and Paul Voss.**

Thanks for this hint. We have reformulated the respective sentence to:

*"While there is some potential for true Lagrangian observations that follow advecting airmasses (Roberts et al. 2016), our quasi-Lagrangian approach provides a similar type of information and can be accomplished via carefully planned aircraft observations."*

We have added the corresponding reference (Roberts et al. 2016).

**References**

Cuxart J., Holtslag, A. A. M., Beare, R., Beljaars, A., Cheng, A., Conangla, L., Ek, M., Freedman, F., Hamdi, R., Kerstein, A., Kitagawa, H., Lenderik, G., Lewellen. D., Mailhot, J., Mauritsen, T., Perov, V., Schayes, G., Steeneveld, G.-J., Svensson, G., Taylor, P., Wunsch, S., Weng, W., and Xu, K.-M. (2006). Single-column intercomparison for a stably stratified atmospheric boundary layer, Bound. Layer Meteorol., 118, 273–303.

Tjernström, M., Sedlar, J., and Shupe, M. D. (2008). How well do regional climate models reproduce radiation and clouds in the Arctic? An evaluation of ARCMIP simulations. J. Appl. Meteorol. Climatol., 47, 2405–2422.

These two references have been added.

**Summary:**

This manuscript closely examines airmass transformations in both warm air outbreak (WAI) and cold air outbreak (CAO) events during the HALO-AC3 campaign. The introduction is lengthy, but very well structured and both appropriately and clearly motivates the specific science questions addressed in this manuscript. A quasi-Lagrangian sampling strategy is employed for sampling WAI and CAO events, with over 2.2 million trajectories calculated for each HALO flight. The caveats of this quasi-Lagrangian approach are clearly defined and discussed in the text (for example: that a true Lagrangian experiment is not possible for aircraft measurements). The figures throughout the manuscript are highly descriptive and very easy (in my view) for the casual reader to understand quickly. ICON does not reproduce the near-surface temperature inversion and has a cold bias below 1 km. ICON specific humidity near the surface is slightly drier compared to the dropsonde data. For the WAI case, ICON performed better over the sea ice than over the open ocean, with MBL evolution well captured. For the CAO case, the lack of a properly simulated inversion in ICON may explain why the difference between the dropsonde & ICON temperatures jumps from a -4K cold bias to a +2K warm bias. I agree with the conclusion that the simulations produce both events accurately, despite noted biases in temperature/specific humidity as well as the differences likely caused by simplified microphysics. I also agree with the conclusion that the change rates for the WAI case are representative of the WAI dataset (from Table 2). The CAO case is representative too – with noted overestimation of the rate of change of RH increase due in part to an overestimation of the heating rate & underestimation of moistening rate near the surface. WAI change rates, overall, were not accurately captured in the simulations. One of the key findings in this manuscript was the dominance of latent heating and turbulence compared to radiative effects during CAO events. For both CAOs and WAIs, adiabatic processes dominated along trajectories for both process rates of temperature & humidity.

One aspect of this manuscript that I appreciate is the relation to previous & related studies on the subject matter (e.g., L388-393 comparing present temperature & humidity change rates to those results found in Kahnert et al. 2021). The authors throughout the manuscript demonstrate up-to-date knowledge on the topic of WAIs and CAOs, and convey the latest set of articles in a way that (in my opinion) is very friendly to the non-Arctic weather or climate researcher who may read this paper. This manuscript is exceptionally well written, organized, and clear – not an easy task given the length and density of this manuscript. The manuscript represents a very important and timely contribution as well given the broader community's interest in WAIs and CAOs on high-latitude climate. I have nothing substantive to add or opinionate on with regards to suggested revisions, aside from a couple very minor comments listed below. In my view, the manuscript is ready for publication.

Thank you very much for this overall positive evaluation.

**General Comments:**

**Somewhere in the last two paragraphs of your Introduction, I recommend explicitly listing Objectives 1-3 numerically. This will make it easier for the reader to hop back-and-forth in the paper, so they are fully clear which data/methods address their respective objective(s).**

This idea makes sense. We have implemented a list of the three objectives in the second last paragraph of the Introduction. Here is the new text:

"… we pursue three objectives in this paper:

- *Objective 1: We test the ability of the ICON model to reproduce measurements of vertical profile of thermodynamic and cloud quantities from dropsondes and cloud radar in an Eulerian framework. First, two specific cases are used to showcase our approach: a massive WAI (13 March 2022), and a pronounced CAO (01 April 2022). Secondly, the Eulerian measurement-model comparisons are extended to results from further cases from flights over the entire measurement period (six days with WAIs, six days with CAOs).*
- *Objective 2: We exploit the ICON simulations to investigate the thermodynamic and cloud evolution of the airmasses along their trajectories. This enables to study the role of adiabatic versus diabatic processes for temperature changes, which is further refined to the specific diabatic effects of radiation, latent heat, and turbulence.*
- *Objective 3: We conduct a novel quasi-Lagrangian model evaluation by testing how well the ICON model simulates measured heating and cooling rates (temperature change rates), as well as moistening and drying rates (humidity change rates)."*

**Specific Comments:**

**L108: (objective 2) I could consider listing the objectives numerically (1., 2., and 3.) somewhere in the introduction, especially since this 2$^{nd}$ objective is discussed before the 1$^{st}$ objective.**

We have rearranged the order in which the three objectives are mentioned. The last paragraph of the Introduction (Section 1) clearly relates the objectives to the corresponding sections. The changed paragraph reads:

*"This article is structured in six sections. After the introduction (Section 1), Section 2 describes the simulations, measurements, as well as the Eulerian and quasi-Lagrangian sampling strategies applied in this study. As the quasi-Lagrangian approach heavily relies on the quality of trajectories, their quality is assessed in Appendix A. The three main parts (Sections 3-5) address the three objectives of the paper. They contain the Eulerian comparisons of ICON model results with aircraft observations collected during WAIs and CAOs (Section 3, Objective 1), and the discussion of modeled airmass transformations and processes driving them (Section 4, Objective 2). Section 5 discusses ICON model results and the corresponding measurements quantifying the temperature and humidity change rates during transport of*

*airmasses (Objective 3). The final part of this paper (Section 6) summarizes the discussion and concludes the article."*

**L120-125: This paragraph highlights dropsondes, but the dropsonde variables are not listed/discussed until L192. Consider briefly describing in a sentence or two what data the dropsondes provide. Since dropsondes are central to the analysis, I would also recommend adding some basic information about what sensors are contained in each dropsondes, manufacturer information, etc. and provide an additional appropriate reference or two.**

We have added some more information on the dropsonde variables that might be helpful to the reader. We have included the following text:

*"The deployed RD41 dropsondes measured air pressure (accuracy: 0.4 hPa), T (accuracy: 0.1 K), RH (accuracy: 2 %), as well as horizontal wind speeds derived from a Global Positioning System (GPS) receiver (accuracy: 0.2 m $s^{-1}$) (Vaisala, 2020, Ehrlich et al. 2025}. $\Theta_e$ and q were derived from the measured parameters."*

**L135: The average reader may not know what a "moist tongue" is. I personally think it's quite descriptive & implies what it means, but it may be worth adding a parenthetical clarification (referring to the region of 200 kg/(m s) IVT?).**

We have deleted the not well-defined term "moisture tongue" and replaced it by "core of this WAI". The revised sentence reads:

*"The flight transected through the core of this WAI at around 75 °N in the Fram Strait until crossing the sea ice edge and continued northward with a total of seven transects of the moist airmass."*

**L142: "with only sea ice being present at latitudes higher than 80N"... this is a bit misleading, as Figure 1 on the next page shows the mean sea ice concentrations at latitudes south of 80N, albeit in regions where fewer of the flight measurements took place.**

We have deleted this misleading part of the sentence.

**L245: Forgot "diagram" after "so-called contoured frequency by altitude".**

We have included the missing "diagram".

**L315: This is a great note to add to the manuscript, but I think it may be worth repeating once in Figure 6 (i.e., that the subset is chosen specifically to improve clarity/reduce overcrowding in the plot).**

We have deleted this note from the text and included it into the figure captions of Figs. 6, 7, and 9 to avoid confusion of the reader as noted by Reviewer #1.

**L371: "wider" atmospheric layers?**

We have deleted the "wider".

**Paper summary:**

This paper evaluates the performance of a NWP model, ICON in limited-area form, through Eulerian and quasi-Lagrangian comparisons against measurements during airmass transformation periods (both WAIs and CAOs) in the Arctic, specifically over the Norwegian Sea and the Arctic sea ice. The focus is on profiles of thermodynamic and cloud properties. The measurements were made aboard an aircraft, which sampled the same air parcels multiple times. That allows observationally-based estimation of quasi-Lagrangian change (i.e., the rate of airmass modification through turbulent and radiative fluxes), that can be compared against matching trajectories from the ICON simulation.

**Overall evaluation:**

This is a thorough evaluation of a model's performance in capturing Arctic airmass modification in both directions. The analysis is rigorous, the illustrations are in-depth, and the results on the roles of adiabatic and diabatic processes at the surface and in the atmosphere robust. The story across the paper is excellent, starting with a broad-brush up to date Introduction of previous work, followed by enough depth to address the findings and limitations of the figures shown, and at the same time enough vision from above not to be mired in details but rather focus on the main take-aways.

**General comments:**

1. **The paper could build a stronger rationale for the focus on periods of rapid airmass transformation and change. A critic could argue that such periods should be avoided because much of the model-observation differences can be due to initial condition uncertainty (because of the rapid change and the poor time resolution of the model's driver dataset, i.e. the operational global ICON initialized at 00Z)? A lot of campaigns and studies have focused on more balanced Arctic conditions, when model deficiencies usually are more persistent and model biases are more attributable to cloud, radiative, or surface exchange processes. But the HALO-AC3 flight campaign and this study expressly targeted periods of rapid change, precisely to understand how well models can capture such change. Still, initial condition uncertainty remains and the paper hardly addresses this "noise" source.**

   Initial conditions as well as lateral boundary conditions are an issue for almost all model simulations and comparisons. We picked explicitly situations with rapid changes due to their strong influence on the environment and in order to evaluate their representation in NWP simulations. By running higher resolution simulations, the model results get actually comparable to observations and a more detailed evaluation will be possible then it would have been for using the global icon forecast directly. But of course, there are some well known parts – e.g. the representation of the boundary layer above sea ice

– which are inherited from the global forcing model. This influence was not in the focus of the paper, but would be a study on its own.

2. **It is not clear how this paper builds on and is different from other recent papers led by the same lead author and mentioned in the Introduction, (Wendisch et al. 2023a, b, Wendisch et al. 2024)**

This comment was also made by Reviewer #1. Here we copy our reply from above:

The Wendisch et al. (2023a) paper is a general introduction into the $(\mathcal{AC})^3$ project and its first results, it mentions the HALO –$(\mathcal{AC})^3$ campaign but provides no detailed data analysis of the corresponding measurements. It should be noted that the $(\mathcal{AC})^3$ project involves much more than the HALO –$(\mathcal{AC})^3$ campaign, which is only one of the many activities within $(\mathcal{AC})^3$. The Wendisch et al. (2023b) paper considers one specific aspect of the influence of airmass transformations, namely the impact of surface properties on the radiative energy budget. That has not been investigated in the current manuscript. Furthermore, no data from the HALO –$(\mathcal{AC})^3$ campaign have been used in Wendisch et al. (2023b). The focus of the Wendisch et al. (2024) paper is the general introduction of the HALO –$(\mathcal{AC})^3$ campaign. In this paper the quasi-Lagrangian strategy is introduced, but only exemplarily measurements are shown. In the current manuscript we compare measured and ICON-simulated change rates of thermodynamic and cloud properties. In this regard, all cited papers cover different areas with only minimal overlapping in the description of the motivation and the methodology of the quasi-Lagrangian approach, which appear needed for stand-alone papers.

To clearly indicate the distinctions between the cited Wendisch et al. (2024) and the current submission, we have replaced the last two sentences of the second-last paragraph of the introductory section by:

*"HALO-$(\mathcal{AC})^3$ delivered numerous observations of thermodynamic and cloud properties along pronounced WAIs and CAOs over open ocean and sea ice, which have been introduced and summarized by Wendisch et al. (2024). This publication also motivated extensively the general need for a Lagrangian-based model evaluation and the required quasi-Lagrangian observations, including their practical realization by aircraft measurements. In the current study, we go one step beyond by exploiting the HALO-$(\mathcal{AC})^3$ measurements in synergy with simulations conducted with the ICON (Icosahedral Nonhydrostatic) weather forecast model to investigate airmass transformations during WAIs and CAOs."*

3. like key results listed clearly, like a set of bullets. The long paragraph starting on L451 ("The observational and modeling results …") lends itself well to a break-up in a specific list.

We have separated this long paragraph into several parts, as also suggested by Reviewer #2.

**Specific comments, typos, and technical issues:**

**Line 148: I suggest interpolating the hourly model data to the time of the drop sonde, like done later in the paper.**

Thanks for identifying this obvious error. The procedure suggested by the reviewer is actually what we have done already in the original manuscript and how the comparisons between dropsonde measurements and model results have been realized. The original sentence is incorrect in this regard. Sorry for this mistake. More specifically, the hourly model output was linearly interpolated to 1 min resolution, to match the temporal resolution of the PAMTRA simulations and to be much closer in time to the measurements (reference time again being when the dropsonde reaches ground level).

We have replaced the original text:

*"… the hourly model output was linearly interpolated to 1 min resolution, to match the temporal resolution of the PAMTRA simulations and to be much closer in time to the measurements."*

**Line 230: The dropsonde drift is mentioned but not accounted for. "As the dropsonde is traveling in space, while the model column is constant, this introduces some uncertainty, especially in highly variable situations, such as the MIZ." (Lines 213-215). This drift can easily be accounted for. Rather than using the dropsonde's final location (where the sonde hit the Earth surface), why not compare geographically more precise profiles, using the actual GPS location of the sonde at each level? Anyway, the dropsonde should not drift more than a couple of model grid points, so this uncertainty should be small.**

We always took the location of the dropsonde at its lowest point above ground as reference ($z \approx 0$ km). In this way the atmospheric boundary layer (with strong interactions with the underlying surface) is well accounted for. Furthermore, horizontal wind speeds were generally below 25 m s$^{-1}$ (Fig. A1). At a typical dropsonde descent rate of 11 m s$^{-1}$ (Vaisala 2020), a vertical drop of 1 km took the dropsonde around 90 seconds. This corresponds to a maximum horizontal drift of 2.3 km, or slightly less than the width of one grid cell (2.4 km). The lowest 2 km of the dropsonde descent thus corresponds to only two grid cells.

We have added the following text correspondingly:

*"Please note that the horizontal drift of the dropsondes during their vertical fall, which was always less than 30 km from release at HALO flight altitude to touchdown on the ground, was not taken into account. Considering the horizontal wind speeds, which were generally below 25 m s$^{-1}$ (Fig. A1), and the typical dropsonde descent rate of 11 m s$^{-1}$ (Vaisala, 2020), a vertical fall of 1 km takes the dropsonde around 90 seconds. This corresponds to a maximum horizontal drift of 2.3 km, which is slightly less than the width of one ICON model grid cell (2.4 km). If the dropsonde falls 2 km vertically, it drifts horizontally through only two grid cells, which should not significantly bias the Eulerian measurement-model comparison. Furthermore, the hourly model output was linearly interpolated to 1 min resolution, to match the temporal resolution of the PAMTRA simulations and to be much closer in time to the measurements."*

**Line 313: Add 'change': "Lastly, the magnitude of humidity change rates are compared ... "**

Done.

**Line 319: Fig. 6 suggests that most cloud ice is below 3 km altitude, mostly dry air above 3.5 km altitude for the WAI. The next sentence mentions correctly that most solid precipitation is below 3 km.**

Thanks, this mistake has been corrected in the text, which reads in revised form:

*"There is relatively little cloud ice, with most at altitudes _below_ 3 km."*

**Line 330: Typo: The time-series of the 1-hourly ...**

Corrected.

**Line 380: How can there be sub-hour variability in some of the variables when it is a linear interpolation between hourly data points?**

That is a very careful observation. The sub-hourly variability seen in Figs. 7-9 results from the fact that trajectories are moving through both time and space. Model output was available every hour, but most trajectories traversed several model grid points within each one-hour time window. The variability as seen in the plots thus represents the combined temporal plus spatial variability along the trajectories.